# Understanding Certified Training with Interval Bound Propagation

**Yuhao Mao, Mark Niklas Müller, Marc Fischer & Martin Vechev**
Department of Computer Science, ETH Zürich, Swizterland
`{yuhao.mao, mark.mueller, marc.fischer, martin.vechev}@inf.ethz.ch`

## Abstract

As robustness verification methods are becoming more precise, training certifiably robust neural networks is becoming ever more relevant. To this end, certified training methods compute and then optimize an upper bound on the worst-case loss over a robustness specification. Curiously, training methods based on the imprecise interval bound propagation (IBP) consistently outperform those leveraging more precise bounds. Still, we lack a theoretical understanding of the mechanisms making IBP so successful. In this work, we investigate these mechanisms by leveraging a novel metric measuring the tightness of IBP bounds. We first show theoretically that, for deep linear models (DLNs), tightness decreases with width and depth at initialization, but improves with IBP training. We, then, derive sufficient and necessary conditions on weight matrices for IBP bounds to become exact and demonstrate that these impose strong regularization, providing an explanation for the observed robustness-accuracy trade-off. Finally, we show how these results on DLNs transfer to ReLU networks, before conducting an extensive empirical study, (i) confirming this transferability and yielding state-of-the-art certified accuracy, (ii) finding that while all IBP-based training methods lead to high tightness, this increase is dominated by the size of the propagated input regions rather than the robustness specification, and finally (iii) observing that non-IBP-based methods do not increase tightness. Together, these results help explain the success of recent certified training methods and may guide the development of new ones.

## 1 Introduction

The increasing deployment of deep-learning-based systems in safety-critical domains has made their trustworthiness and especially formal robustness guarantees against adversarial examples (Biggio et al., 2013; Szegedy et al., 2014) an ever more important topic. As significant progress has been made on neural network certification (Zhang et al., 2022; Ferrari et al., 2022), the focus in the field is increasingly shifting to the development of specialized training methods that improve certifiable robustness while minimizing the accompanying reduction in standard accuracy.

**Certified Training**   These certified training methods aim to compute and then optimize approximations of the network's worst-case loss over an input region defined by an adversary specification. To this end, they compute an over-approximation of the network's reachable set using symbolic bound propagation methods (Singh et al., 2018; 2019b; Gowal et al., 2018). Surprisingly, training methods based on the least precise bounds, obtained via interval bound propagation (IBP), empirically yield the best performance (Shi et al., 2021). Jovanović et al. (2022) investigate this surprising observation theoretically and find that more precise bounding methods induce harder optimization problems.

As a result, *all* methods obtaining state-of-the-art performance leverage IBP bounds either directly (Shi et al., 2021), as regularizer (Palma et al., 2022), or to precisely but unsoundly approximate the worst-case loss (Müller et al., 2022b; Mao et al., 2023; Palma et al., 2023). However, while IBP is crucial to their success, none of these works develop a theoretical understanding of what makes IBP training so effective and how it affects bound tightness and network regularization.

**This Work**   We take a first step towards building a deeper understanding of the mechanisms making IBP training so successful and thereby pave the way for further advances in certified training. To this end, we derive necessary and sufficient conditions on a network's weights under which IBP

bounds become tight, a property we call *propagation invariance*, and prove that it implies an extreme regularization, agreeing well with the empirically observed trade-off between certifiable robustness and accuracy (Tsipras et al., 2019; Müller et al., 2022b). To investigate how close real networks are to full propagation invariance, we introduce the metric *propagation tightness* as the ratio of optimal and IBP bounds, and show how to efficiently compute it globally for deep linear networks (DLNs) and locally for ReLU networks.

This novel metric enables us to theoretically investigate the effects of model architecture, weight initialization, and training methods on IBP bound tightness for deep linear networks (DLNs). We show that (i) at initialization, tightness decreases with width (polynomially) and depth (exponentially), (ii) tightness is increased by IBP training, and (iii) sufficient width becomes crucial for trained networks.

Conducting an extensive empirical study, we confirm the predictiveness of our theoretical results for deep ReLU networks and observe that: (i) increasing network width but not depth improves state-of-the-art certified accuracy, (ii) IBP training significantly increases tightness, almost to the point of propagation invariance, (iii) unsound IBP-based training methods increase tightness to a smaller degree, determined by the size of the propagated input region and the weight of the IBP-loss, but yield better performance, and (iv) non-IBP-based training methods barely increase tightness, leading to higher accuracy but worse robustness. These findings suggest that while IBP-based training methods improve robustness by increasing tightness at the cost of standard accuracy, high tightness is not generally necessary for robustness. This observation explains the recent success of unsound IBP-based methods and, in combination with the theoretical and practical insights developed here, promises to be a key step toward constructing novel and more effective certified training methods.

## 2  BACKGROUND

Here, we provide a background on adversarial and certified robustness. We consider a classifer $\boldsymbol{f} \colon \mathbb{R}^{d_{\text{in}}} \mapsto \mathbb{R}^c$ predicting a numerical score $\boldsymbol{y} := \boldsymbol{f}(\boldsymbol{x})$ per class given an input $\boldsymbol{x} \in \mathcal{X} \subseteq \mathbb{R}^{d_{\text{in}}}$.

**Adversarial Robustness** describes the property of a classifer $\boldsymbol{f}$ to consistently predict the target class $t$ for all perturbed inputs $\boldsymbol{x}'$ in an $\ell_p$-norm ball $\mathcal{B}_p^{\epsilon_p}(\boldsymbol{x})$ of radius $\epsilon_p$. As we focus on $\ell_\infty$ perturbations in this work, we henceforth drop the subscript $p$ for notational clarity. More formally, we define *adversarial robustness* as:

$$\arg\max_j f(\boldsymbol{x}')_j = t, \quad \forall \boldsymbol{x}' \in \mathcal{B}_p^{\epsilon_p}(\boldsymbol{x}) := \{\boldsymbol{x}' \in \mathcal{X} \mid \|\boldsymbol{x} - \boldsymbol{x}'\|_p \le \epsilon_p\}. \tag{1}$$

**Neural Network Certification** can be used to formally prove the robustness of a classifier $\boldsymbol{f}$ for a given input region $\mathcal{B}^\epsilon(\boldsymbol{x})$. Interval bound propagation (IBP) (Gowal et al., 2018; Mirman et al., 2018) is a simple but popular such certification method. It is based on propagating an input region $\mathcal{B}^\epsilon(\boldsymbol{x})$ through a neural network by computing Box over-approximations (each dimension is described as an interval) of the hidden state after every layer until we reach the output space. There, it is checked whether all points in the resulting over-approximation of the network's reachable set yield the correct classification. As an example, consider an $L$-layer network $\boldsymbol{f} = \boldsymbol{h}_L \circ \boldsymbol{\sigma} \circ \boldsymbol{h}_{L-2} \circ \ldots \circ \boldsymbol{h}_1$, with linear layers $\boldsymbol{h}_i$ and ReLU activation functions $\boldsymbol{\sigma}$. We first over-approximate the input region $\mathcal{B}^\epsilon(\boldsymbol{x})$ as Box with radius $\boldsymbol{\delta}^0 = \epsilon$ and center $\dot{\boldsymbol{x}}^0 = \boldsymbol{x}$, such that we have the $i^{\text{th}}$ dimension of the input $x_i^0 \in [\underline{x}_i, \bar{x}_i] := [\dot{x}_i^0 - \delta_i^0, \dot{x}_i^0 + \delta_i^0]$. Propagating such a Box through the linear layer $\boldsymbol{h}_i(\boldsymbol{x}^{i-1}) = \boldsymbol{W}\boldsymbol{x}^{i-1} + \boldsymbol{b} =: \boldsymbol{x}^i$, we obtain the output hyperbox with centre $\dot{\boldsymbol{x}}^i = \boldsymbol{W}\dot{\boldsymbol{x}}^{i-1} + \boldsymbol{b}$ and radius $\boldsymbol{\delta}^i = |\boldsymbol{W}|\boldsymbol{\delta}^{i-1}$, where $|\cdot|$ denotes the element-wise absolute value. To propagate a Box through the ReLU activation $\text{ReLU}(\boldsymbol{x}^{i-1}) := \max(0, \boldsymbol{x}^{i-1})$, we propagate the lower and upper bound separately, resulting in an output Box with $\dot{\boldsymbol{x}}^i = \frac{\bar{\boldsymbol{x}}^i + \boldsymbol{x}^i}{2}$ and $\boldsymbol{\delta}^i = \frac{\bar{\boldsymbol{x}}^i - \boldsymbol{x}^i}{2}$ where $\boldsymbol{x}^i = \text{ReLU}(\dot{\boldsymbol{x}}^{i-1} - \boldsymbol{\delta}^{i-1})$ and $\bar{\boldsymbol{x}}^i = \text{ReLU}(\dot{\boldsymbol{x}}^{i-1} + \boldsymbol{\delta}^{i-1})$. We proceed this way for all layers obtaining first lower and upper bounds on the network's output $\boldsymbol{y}$ and then an upper bound $\bar{\boldsymbol{y}}^\Delta$ on the logit difference $y_i^\Delta := y_i - y_t$. Showing that $\bar{y}_i^\Delta < 0, \ \forall i \ne t$ is then equivalent to proving adversarial robustness on the considered input region.

We illustrate this propagation process for a two-layer network in Figure 1. There, we show the exact propagation of the input region ■ in blue, its optimal box approximation ■ in green, and the IBP approximation as dashed boxes ⊡. Note how after the first linear and ReLU layer (third column), the box approximations (both optimal ■ and IBP ⊡) contain already many points outside the reachable set

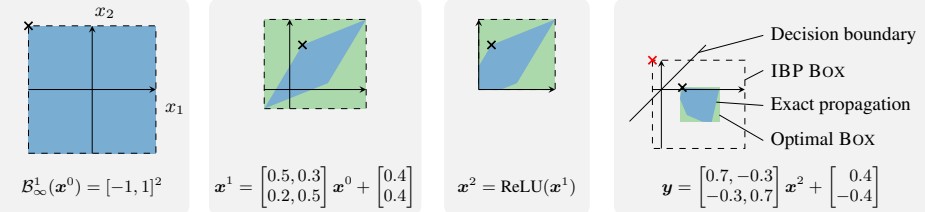

Figure 1: Comparison of exact (■), optimal box (■), and IBP (⬚) propagation through a one layer network. We show the concrete points maximizing the logit difference $y_2 - y_1$ as a black × and the corresponding relaxation as a red ×.

■, despite it being the smallest hyper-box containing the exact region. These so-called approximation errors accumulate quickly when using IBP, leading to an increasingly imprecise abstraction, as can be seen by comparing the optimal box ■ and IBP ⬚ approximation after an additional linear layer (rightmost column). To verify that this network classifies all inputs in $[-1, 1]^2$ to class 1, we have to show the upper bound of the logit difference $y_2 - y_1$ to be less than 0. While the concrete maximum of $-0.3 \geq y_2 - y_1$ (black ×) is indeed less than 0, showing that the network is robust, IBP ⬚ only yields $0.6 \geq y_2 - y_1$ (red ×) and is thus too imprecise to prove it. In contrast, the optimal box ■ yields a precise approximation of the true reachable set, sufficient to prove robustness.

**Training for Robustness** is required to obtain (certifiably) robust neural networks. For a data distribution $(\boldsymbol{x}, t) \sim \mathcal{D}$, standard training optimizes the network parametrization $\boldsymbol{\theta}$ to minimize the expected cross-entropy loss:

$$\theta_{\text{std}} = \arg \min_{\theta} \mathbb{E}_{\mathcal{D}}[\mathcal{L}_{\text{CE}}(\boldsymbol{f_\theta}(\boldsymbol{x}), t)], \quad \text{with} \quad \mathcal{L}_{\text{CE}}(\boldsymbol{y}, t) = \ln \big(1 + \sum_{i \neq t} \exp(y_i - y_t)\big). \quad (2)$$

To train for robustness, we, instead, aim to minimize the expected *worst-case loss* for a given robustness specification, leading to a min-max optimization problem:

$$\theta_{\text{rob}} = \arg \min_{\theta} \mathbb{E}_{\mathcal{D}} \left[ \max_{\boldsymbol{x}' \in \mathcal{B}^\epsilon(\boldsymbol{x})} \mathcal{L}_{\text{CE}}(\boldsymbol{f_\theta}(\boldsymbol{x}'), t) \right]. \quad (3)$$

As computing the worst-case loss by solving the inner maximization problem is generally intractable, it is commonly under- or over-approximated, yielding adversarial and certified training, respectively.

**Adversarial Training** optimizes a lower bound on the inner optimization objective in Equation (3). It first computes concrete examples $\boldsymbol{x}' \in \mathcal{B}^\epsilon(\boldsymbol{x})$ that approximately maximize the loss term $\mathcal{L}_{\text{CE}}$ and then optimizes the network parameters $\boldsymbol{\theta}$ for these examples. While networks trained this way typically exhibit good empirical robustness, they remain hard to formally certify and are sometimes vulnerable to stronger attacks (Tramèr et al., 2020; Croce & Hein, 2020).

**Certified Training** typically optimizes an upper bound on the inner maximization objective in Equation (3). The resulting robust cross-entropy loss $\mathcal{L}_{\text{CE,rob}}(\mathcal{B}^\epsilon(\boldsymbol{x}), t) = \mathcal{L}_{\text{CE}}(\overline{\boldsymbol{y}}^\Delta, t)$ is obtained by first computing an upper bound $\overline{\boldsymbol{y}}^\Delta$ on the logit differences $\boldsymbol{y}^\Delta := \boldsymbol{y} - y_t$ with a bound propagation method as described above and then plugging it into the standard cross-entropy loss.

Surprisingly, the imprecise IBP bounds (Mirman et al., 2018; Gowal et al., 2018; Shi et al., 2021) consistently yield better performance than methods based on tighter approximations (Wong et al., 2018; Zhang et al., 2020; Balunovic & Vechev, 2020). Jovanović et al. (2022) trace this back to the optimization problems induced by the more precise methods becoming intractable to solve.

However, the heavy regularization that makes IBP trained networks amenable to certification also severely reduces their standard accuracy. To alleviate the resulting robustness-accuracy trade-off, *all* current state-of-the-art certified training methods combine IBP and adversarial training by using IBP bounds only for regularization (IBP-R (Palma et al., 2022)), by only propagating small, adversarially selected regions (SABR (Müller et al., 2022b)), using IBP bounds only for the first layers and PGD bounds for the remainder of the network (TAPS (Mao et al., 2023)), or combining losses over adversarial samples and IBP bounds (CC-IBP, MTL-IBP (Palma et al., 2023)).

In light of this surprising dominance of IBP-based training methods, understanding the regularization IBP induces and its effect on tightness promises to be a key step towards developing novel and more effective certified training methods.

## 3 UNDERSTANDING IBP TRAINING

In this section, we theoretically investigate the relationship between the box bounds obtained by layer-wise propagation, *i.e.*, IBP, and optimal propagation. We illustrate both in Figure 1 and note that the latter are sufficient for exact robustness certification (see Lemma 3.1). First, we formally define layer-wise (IBP) and optimal box propagation, before deriving sufficient and necessary conditions under which the resulting bounds become identical. Then, we show that these conditions induce strong regularization, motivating us to introduce the propagation tightness $\tau$ as a relaxed measure of bound precision, which can be efficiently computed globally for deep linear (DLN) and locally for ReLU networks. Based on these results, we first investigate how tightness depends on network architecture at initialization, before showing that it improves with IBP training. Finally, we demonstrate that even linear dimensionality reduction is inherently imprecise for both optimal and IBP propagation, making sufficient network width key for tight box bounds. We defer all proofs to App. B.

**Setting** We focus our theoretical analysis on deep linear networks (DLNs), *i.e.*, $\boldsymbol{f}(x) = \Pi_{i=1}^{L} \boldsymbol{W}^{(i)} \boldsymbol{x}$, popular for theoretical discussion of neural networks (Saxe et al., 2014; Ji & Telgarsky, 2019; Wu et al., 2019). While such a reduction of a ReLU network to an overall linear function may seem restrictive, it preserves many interesting properties and allows for theoretical insights, while ReLU networks are theoretically unwieldy. As ReLU networks become linear for fixed activation patterns, the DLN approximation becomes exact for robustness analysis at infinitesimal perturbation magnitudes. Further, DLNs retain the layer-wise structure and joint non-convexity in the weights of different layers of ReLU networks, making them a widely popular analysis tool (Ribeiro et al., 2016). After proving key results on DLNs, we will show how they transfer to ReLU networks.

### 3.1 LAYER-WISE AND OPTIMAL BOX PROPAGATION

We define the optimal hyper-box approximation $\text{Box}^*(\boldsymbol{f}, \mathcal{B}^\epsilon(\boldsymbol{x}))$ as the smallest hyper-box $[\underline{\boldsymbol{z}}, \overline{\boldsymbol{z}}]$ such that it contains the image $\boldsymbol{f}(\boldsymbol{x}')$ of all points $\boldsymbol{x}'$ in $\mathcal{B}^\epsilon(\boldsymbol{x})$, *i.e.*, $\boldsymbol{f}(\boldsymbol{x}') \in [\underline{\boldsymbol{z}}, \overline{\boldsymbol{z}}], \forall \boldsymbol{x}' \in \mathcal{B}^\epsilon(\boldsymbol{x})$. Similarly, we define the layer-wise box approximation as the result of sequentially applying the optimal approximation to every layer individually: $\text{Box}^\dagger(\boldsymbol{f}, \mathcal{B}^\epsilon(\boldsymbol{x})) := \text{Box}^*(\boldsymbol{W}_L, \text{Box}^*(\cdots, \text{Box}^*(\boldsymbol{W}^{(1)}, \mathcal{B}^\epsilon(\boldsymbol{x}))))$. We write their upper- and lower-bounds as $[\underline{\boldsymbol{z}}^*, \overline{\boldsymbol{z}}^*]$ and $[\underline{\boldsymbol{z}}^\dagger, \overline{\boldsymbol{z}}^\dagger]$, respectively. We note that optimal box bounds on the logit differences $\boldsymbol{y}^\Delta := \boldsymbol{y} - y_t$ (instead of on the logits $\boldsymbol{y}$ as shown in Figure 1) are sufficient for exact robustness verification:

**Lemma 3.1.** *Any $\mathcal{C}^0$ continuous classifier $\boldsymbol{f}$, computing the logit difference $y_i^\Delta := y_i - y_t, \forall i \neq t$, is robustly correct on $\mathcal{B}^\epsilon(\boldsymbol{x})$ if and only if $\text{Box}^*(\boldsymbol{f}, \mathcal{B}^\epsilon(\boldsymbol{x})) \subseteq \mathbb{R}_{<0}^{c-1}$, i.e., $\bar{y}_i^{\Delta^*} < 0, \forall i \neq t$.*

For DLNs, we can efficiently compute both optimal $\text{Box}^*$ and layerwise $\text{Box}^\dagger$ box bounds as follows:

**Theorem 3.2** (Box Propagation)**.** *For an $L$-layer DLN $\boldsymbol{f} = \Pi_{k=1}^{L} \boldsymbol{W}^{(k)}$, we obtain the box centres $\dot{\boldsymbol{z}}^* = \dot{\boldsymbol{z}}^\dagger = \boldsymbol{f}(\boldsymbol{x})$ and the radii*

$$\frac{\overline{\boldsymbol{z}}^* - \underline{\boldsymbol{z}}^*}{2} = \left| \Pi_{k=1}^{L} \boldsymbol{W}^{(k)} \right| \boldsymbol{\epsilon}, \quad and \quad \frac{\overline{\boldsymbol{z}}^\dagger - \underline{\boldsymbol{z}}^\dagger}{2} = \left( \Pi_{k=1}^{L} \left| \boldsymbol{W}^{(k)} \right| \right) \boldsymbol{\epsilon}. \tag{4}$$

Comparing the radius computation of the optimal and layer-wise approximations, we observe that the main difference lies in where the element-wise absolute value $|\cdot|$ of the weight matrix is taken. For the optimal box, we first multiply all weight matrices before taking the absolute value $|\Pi_{k=1}^{L} \boldsymbol{W}^{(k)}|$, thus allowing for cancellations of terms of opposite signs. For the layer-wise approximation, in contrast, we first take the absolute value of each weight matrix before multiplying them together $\Pi_{k=1}^{L} |\boldsymbol{W}^{(k)}|$, thereby losing all relational information between variables. Let us now investigate under which conditions layer-wise and optimal bounds become identical.

### 3.2 PROPAGATION INVARIANCE AND IBP BOUND TIGHTNESS

**Propagation Invariance** We call a network (globally) *propagation invariant* (PI) if the layer-wise and optimal box over-approximations are identical for every input box. Clearly, non-negative weight matrices lead to PI networks (Lin et al., 2022), as the absolute value in Theorem 3.2 loses its effect. However, non-negative weights significantly reduce network expressiveness and performance

(Chorowski & Zurada, 2014), raising the question of whether they are a necessary condition. We show that they are not, by deriving a sufficient *and* necessary condition for a two-layer DLN:

**Lemma 3.3** (Propagation Invariance). *A two-layer DLN $\boldsymbol{f} = \boldsymbol{W}^{(2)}\boldsymbol{W}^{(1)}$ is propagation invariant if and only if for every fixed $(i, j)$, we have $\left|\sum_k W_{i,k}^{(2)} \cdot W_{k,j}^{(1)}\right| = \sum_k |W_{i,k}^{(2)} \cdot W_{k,j}^{(1)}|$, i.e., either $W_{i,k}^{(2)} \cdot W_{k,j}^{(1)} \geq 0$ for all $k$ or $W_{i,k}^{(2)} \cdot W_{k,j}^{(1)} \leq 0$ for all $k$.*

**Conditions for Propagation Invariance**   To see how strict the condition described by Lemma 3.3 is, we observe that propagation invariance requires the sign of the last element in any two-by-two block in $\boldsymbol{W}^{(2)}\boldsymbol{W}^{(1)}$ to be determined by the signs of the other three elements:

**Theorem 3.4** (Non-Propagation Invariance). *Assume $\exists i, i', j, j'$, such that $W_{\cdot,j}^{(1)}$, $W_{\cdot,j'}^{(1)}$, $W_{i,\cdot}^{(2)}$ and $W_{i',\cdot}^{(2)}$ are all non-zero. If $(\boldsymbol{W}^{(2)}\boldsymbol{W}^{(1)})_{i,j} \cdot (\boldsymbol{W}^{(2)}\boldsymbol{W}^{(1)})_{i,j'} \cdot (\boldsymbol{W}^{(2)}\boldsymbol{W}^{(1)})_{i',j} \cdot (\boldsymbol{W}^{(2)}\boldsymbol{W}^{(1)})_{i',j'} < 0$, then $\boldsymbol{f} = \boldsymbol{W}^{(2)}\boldsymbol{W}^{(1)}$ is not propagation invariant.*

To obtain a propagation invariant network with weights $\boldsymbol{W}^{(2)}\boldsymbol{W}^{(1)} \in \mathcal{R}^{d \times d}$, we can thus only choose $2d - 1$ (e.g., one row and one column) of the $d^2$ signs freely (see Corollary A.1 in App. A).

The statements of Lemma 3.3 and Theorem 3.4 naturally extend to DLNs with more than two layers $L > 2$. However, the conditions within Theorem 3.4 become increasingly complex and strict as more and more terms need to yield the same sign. Thus, we focus our analysis on $L = 2$ for clarity.

**IBP Bound Tightness**   To analyze the tightness of IBP bounds for networks that do not satisfy the strict conditions for propagation invariance, we relax it to introduce *propagation tightness* as the ratio between the optimal and layer-wise box radii, simply referred to as *tightness* in this paper.

**Definition 3.5.** *Given a DLN $\boldsymbol{f}$, we define the global propagation tightness $\tau$ as the ratio between optimal $\mathrm{Box}^*(\boldsymbol{f}, \mathcal{B}^\epsilon(\boldsymbol{x}))$ and layer-wise $\mathrm{Box}^\dagger(\boldsymbol{f}, \mathcal{B}^\epsilon(\boldsymbol{x}))$ approximation radius, i.e., $\tau = \frac{\overline{\boldsymbol{z}}^* - \underline{\boldsymbol{z}}^*}{\overline{\boldsymbol{z}}^\dagger - \underline{\boldsymbol{z}}^\dagger}$.*

Intuitively, tightness measures how much smaller the exact dimension-wise $\mathrm{Box}^*$ bounds are, compared to the layer-wise approximation $\mathrm{Box}^\dagger$, thus quantifying the gap between IBP certified and true adversarial robustness. When tightness equals $1$, the network is propagation invariant and can be certified exactly with IBP; when tightness is close to $0$, IBP bounds become arbitrarily imprecise. We highlight that this is orthogonal to the box diameter $\Delta = \overline{\boldsymbol{z}}^\dagger - \underline{\boldsymbol{z}}^\dagger$, considered by Shi et al. (2021).

**ReLU Networks**   The nonlinearity of ReLU networks leads to locally varying tightness and makes the computation of optimal box bounds intractable. However, for infinitesimal perturbation magnitudes $\epsilon$, the activation patterns of ReLU networks remain stable, making them locally linear. We thus introduce a local version of tightness around concrete inputs.

**Definition 3.6.** *For an $L$-layer ReLU network with weight matrices $\boldsymbol{W}^{(k)}$ and activation pattern $\boldsymbol{d}^{(k)}(\boldsymbol{x}) = \mathbb{1}_{\boldsymbol{x}^{(k-1)} > 0} \in \{0, 1\}^{d_k}$ ($1$ for active and $0$ for inactive ReLUs), depending on the input $\boldsymbol{x}$, we define its local tightness as*

$$\tau = \frac{\frac{d}{d\epsilon}(\overline{\boldsymbol{z}}^* - \underline{\boldsymbol{z}}^*)\big|_{\epsilon=0}}{\frac{d}{d\epsilon}(\overline{\boldsymbol{z}}^\dagger - \underline{\boldsymbol{z}}^\dagger)\big|_{\epsilon=0}} = \frac{\left|\Pi_{k=1}^L \mathrm{diag}(\boldsymbol{d}^{(k)})\boldsymbol{W}^{(k)}\right|\mathbf{1}}{\left(\Pi_{k=1}^L \mathrm{diag}(\boldsymbol{d}^{(k)})\left|\boldsymbol{W}^{(k)}\right|\right)\mathbf{1}}.$$

In Definition 3.6, we calculate tightness as the ratio of box size growth rates, evaluated for an infinitesimal input box size $\epsilon$. In this setting, the ReLU network will not have any unstable neurons, making our analysis exact. Only when considering larger perturbation magnitudes will neurons become unstable, making our analysis an approximation of the tightness at that $\epsilon$. However, for the networks and perturbation magnitudes typically considered in the literature, only a very small fraction ($\approx 1\%$) of neurons are unstable (Müller et al., 2022b). To assess the estimation quality of our local tightness, we show its mean relative error compared to the exact tightness computed with MILP for a small CNN3 in Figure 2 for MNIST and in Figure 14 for CIFAR-10. We find that for perturbations smaller than those used during training ($\epsilon \leq 0.05$) relative errors are extremely small ($< 0.5\%$), and only increase slowly after, reaching $2.2\%$ at $\epsilon = 0.1$.

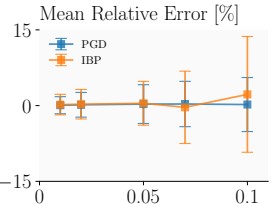

Figure 2: Mean relative error between local tightness (Definition 3.6) and true tightness computed with MILP for a CNN3 trained with PGD or IBP at $\epsilon = 0.05$ on MNIST.

### 3.3 Tightness at Initialization

We first investigate the (expected) tightness $\tau = \frac{\mathbb{E}_{\mathcal{D}_\theta}(\underline{z}^* - \overline{z}^*)}{\mathbb{E}_{\mathcal{D}_\theta}(\overline{z}^\dagger - \underline{z}^\dagger)}$ (independent of the dimension due to symmetry) at initialization, *i.e.*, w.r.t. a weight distribution $\mathcal{D}_\theta$. Let us consider a two-layer DLN at initialization, *i.e.*, with i.i.d. weights following a zero-mean Gaussian distribution $\mathcal{N}(0, \sigma^2)$ with an arbitrary but fixed variance $\sigma^2$ (Glorot & Bengio, 2010; He et al., 2015).

**Lemma 3.7** (Initialization Tightness w.r.t. Width). *Given a 2-layer DLN with weight matrices $\boldsymbol{W}^{(1)} \in \mathcal{R}^{d_1 \times d_0}$, $\boldsymbol{W}^{(2)} \in \mathcal{R}^{d_2 \times d_1}$ with i.i.d. entries from $\mathcal{N}(0, \sigma_1^2)$ and $\mathcal{N}(0, \sigma_2^2)$ (together denoted as $\boldsymbol{\theta}$), we obtain the expected tightness $\tau(d_1) = \frac{\mathbb{E}_\theta(\underline{z}^* - \overline{z}^*)}{\mathbb{E}_\theta(\overline{z}^\dagger - \underline{z}^\dagger)} = \frac{\sqrt{\pi}\,\Gamma(\frac{1}{2}(d_1+1))}{d_1 \Gamma(\frac{1}{2}d_1)} \in \Theta(\frac{1}{\sqrt{d_1}})$.*

Tightness at initialization, thus, decreases quickly with internal width ($\Theta(\frac{1}{\sqrt{d_1}})$), e.g., by a factor of $\tau(500) \approx 0.056$ for the penultimate layer of the popular CNN7 (Gowal et al., 2018; Zhang et al., 2020). It, further, follows directly that tightness will decrease exponentially w.r.t. network depth.

**Corollary 3.8** (Initialization Tightness w.r.t. Depth). *The expected tightness of an L-layer DLN $\boldsymbol{f}$ with minimum internal dimension $d_{min}$ is at most $\tau \leq \tau(d_{min})^{\lfloor \frac{L}{2} \rfloor}$ at initialization.*

This result is independent of the variance $\sigma_1^2, \sigma_2^2$. Thus, tightness at initialization can not be increased by scaling $\sigma^2$, as proposed by Shi et al. (2021) to achieve constant box radius over network depth.

**ReLU Networks** We extend Lemma 3.7 to two-layer ReLU networks (Corollary A.2 in App. A), obtain an expected tightness of $\sqrt{2}\tau(d_1)$, and empirically validate it in Section 4.1.

### 3.4 IBP Training Increases Tightness

We now show theoretically that IBP training increases this tightness. To this end, we again consider a DLN with layer-wise propagation matrix $\boldsymbol{W}^\dagger = \Pi_{i=1}^L |\boldsymbol{W}^{(i)}|$ and optimal propagation matrix $\boldsymbol{W}^* = |\Pi_{i=1}^L \boldsymbol{W}^{(i)}|$, yielding the expected risk for IBP training as $R(\epsilon) = \mathbb{E}_{\boldsymbol{x},y}\mathcal{L}(\text{Box}^\dagger(\boldsymbol{f}, \mathcal{B}^\epsilon(\boldsymbol{x})), y)$.

**Theorem 3.9** (IBP Training Increases Tightness). *Assume homogenous tightness,* i.e., *$\boldsymbol{W}^* = \tau \boldsymbol{W}^\dagger$, and $\frac{\|\nabla_\theta \boldsymbol{W}_{ij}^*\|_2}{\boldsymbol{W}_{ij}^*} \leq \frac{1}{2} \frac{\|\nabla_\theta \boldsymbol{W}_{ij}^\dagger\|_2}{\boldsymbol{W}_{ij}^\dagger}$ for all $i, j$, then, the gradient difference between the IBP and standard loss is aligned with an increase in tightness,* i.e., *$\langle \nabla_\theta(R(\epsilon) - R(0)), \nabla_\theta \tau \rangle \leq 0$ for all $\epsilon > 0$.*

### 3.5 Network Width and Tightness after Training

Many high-dimensional computer vision datasets were shown to have low intrinsic data dimensionality (Pope et al., 2021). Thus, we study the reconstruction loss of a linear embedding into a low-dimensional subspace as a proxy for performance and find that tightness decreases with the width $w$ of a bottleneck layer as long as it is smaller than the data-dimensionality $d$, *i.e.*, $w \ll d$. Further, while reconstruction becomes lossless for points as soon as the width $w$ reaches the intrinsic dimension $k$ of the data, even optimal box propagation requires a width of at least the original data dimension $d$ to achieve loss-less reconstruction. For a $k$-dimensional data distribution, linearly embedded into a $d$ dimensional space with $d \gg k$, the data matrix $X$ has a low-rank eigendecomposition $\text{Var}(X) = U\Lambda U^\top$ with $k$ non-zero eigenvalues. The optimal reconstruction $\hat{X} = U_k U_k^\top X$ is exact by choosing $U_k$ as the $k$ columns of $U$ with non-zero eigenvalues. Yet, box propagation is imprecise:

**Theorem 3.10** (Box Reconstruction Error). *Consider the linear embedding and reconstruction $\hat{\boldsymbol{x}} = U_k U_k^\top \boldsymbol{x}$ of a $d$ dimensional data distribution $\boldsymbol{x} \sim \mathcal{X}$ into a $k$ dimensional space with $d \gg k$ and eigenmatrices $U$ drawn uniformly at random from the orthogonal group. Propagating the input box $\mathcal{B}^\epsilon(\boldsymbol{x})$ layer-wise and optimally, thus, yields $\mathcal{B}^{\delta^\dagger}(\hat{\boldsymbol{x}})$, and $\mathcal{B}^{\delta^*}(\hat{\boldsymbol{x}})$, respectively. Then, we have, (i) $\mathbb{E}(\delta_i/\epsilon) = ck \in \Theta(k)$ for a positive constant $c$ depending solely on $d$ and $c \to \frac{2}{\pi} \approx 0.64$ for large $d$; and (ii) $\mathbb{E}(\delta_i^*/\epsilon) \to \frac{2}{\sqrt{\pi}} \frac{\Gamma(\frac{1}{2}(k+)}{\Gamma(\frac{1}{2}k)} \in \Theta(\sqrt{k})$.*

Intuitively, Theorem 3.10 implies that, while input points can be embedded into and reconstructed from a $k$ dimensional space losslessly, box propagation will yield a box growth of $\Theta(\sqrt{k})$ for optimal and $\Theta(k)$ for layer-wise propagation. However, with $k = d$, we can choose $U_k$ to be an identity matrix, thus obtaining lossless "reconstruction", highlighting the importance of network width.

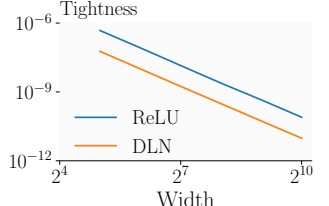 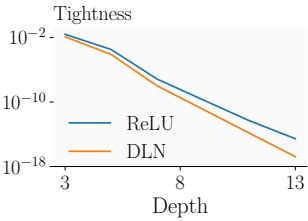 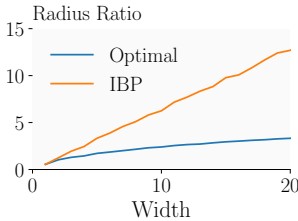

Figure 3: Dependence of tightness at initialization on width (left) and depth (right) for a `CNN7` and CIFAR-10.

Figure 4: Box reconstruction error over bottleneck width $w$.

## 4 EMPIRICAL EVALUATION ANALYSIS

Now, we conduct an empirical study of IBP-based certified training, leveraging our novel tightness metric and specifically its local variant (see Definition 3.6) to gain a deeper understanding of these methods and confirm the applicability of our theoretical analysis to ReLU networks. For certification, we use MN-BAB (Ferrari et al., 2022), a state-of-the-art verifier, and defer further details to App. C.

### 4.1 NETWORK ARCHITECTURE AND TIGHTNESS

First, we confirm the predictiveness of our theoretical results on the effect of network width and depth on tightness at initialization and after training. In Figure 3, we visualize tightness at initialization, depending on network width and depth for DLNs and ReLU networks. As predicted by Lemma 3.7 and Corollary 3.8, tightness decreases polynomially with width (see Figure 3 left) and exponentially with depth (see Figure 3 right), both for DLNs and ReLU networks. We confirm our results on the inherent hardness of linear reconstruction in Figure 4, where we plot the ratio of recovered and original box radii, given a bottleneck layer of width $w$ and synthetic data with intrinsic dimensionality $k = w$. As predicted by Theorem 3.10, IBP propagation yields linear and Box* sublinear growth.

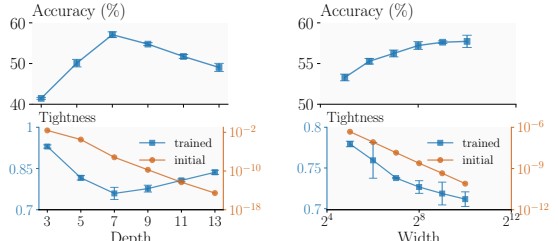

Figure 5: Effect of network depth (top) and width (bottom) on tightness and *training set* IBP-certified accuracy.

Next, we study the interaction of network architecture and IBP training. To this end, we train CNNs with 3 to 13 layers on CIFAR-10 for $\epsilon = 2/255$, visualizing results in Figure 5 (right). To quantify the regularizing effect of propagation tightness, we report training set IBP-certified accuracy as a measure of the goodness of fit. Generally, we would expect increased depth to increase capacity and thus decrease the robust training loss and increase training set accuracy. However, we only observe such an increase in accuracy until a depth of 7 layers before accuracy starts to drop. We can explain this by analyzing the corresponding tightness. As expected, tightness is high for shallow networks but decreases quickly with depth, reaching a minimum for 7 layers. From there, tightness increases again, indicating significant regularization, and thereby decreasing accuracy. This is in line with the popularity of the 7-layer `CNN7` in the certified training literature (Shi et al., 2021; Müller et al., 2022b).

Table 1: Certified and standard accuracy depending on network width.

| Dataset | $\epsilon$ | Method | Width | Accuracy | Certified |
|---|---|---|---|---|---|
| MNIST | 0.1 | IBP | 1× | 98.83 | 98.10 |
| | | | 4× | 98.86 | 98.23 |
| | | SABR | 1× | 98.99 | 98.20 |
| | | | 4× | **98.99** | **98.32** |
| | 0.3 | IBP | 1× | 97.44 | 93.26 |
| | | | 4× | 97.66 | 93.35 |
| | | SABR | 1× | **98.82** | 93.38 |
| | | | 4× | 98.48 | **93.85** |
| CIFAR-10 | $\frac{2}{255}$ | IBP | 1× | 67.93 | 55.85 |
| | | | 2× | 68.06 | 56.18 |
| | | IBP-R | 1× | 78.43 | 60.87 |
| | | | 2× | **80.46** | 62.03 |
| | | SABR | 1× | 79.24 | 62.84 |
| | | | 2× | 79.89 | **63.28** |
| | $\frac{8}{255}$ | IBP | 1× | 47.35 | 34.17 |
| | | | 2× | 47.83 | 33.98 |
| | | SABR | 1× | 50.78 | 34.12 |
| | | | 2× | **51.56** | **34.95** |
| TinyImageNet | $\frac{1}{255}$ | IBP | 0.5× | 24.47 | 18.76 |
| | | | 1× | 25.33 | 19.46 |
| | | | 2× | 25.40 | 19.92 |
| | | SABR | 0.5× | 27.56 | 20.54 |
| | | | 1× | 28.63 | 21.21 |
| | | | 2× | **28.97** | **21.36** |

Continuing our study of architecture effects, we train networks with 0.5 to 16 times the width of a standard `CNN7` using IBP training and visualize the resulting IBP certified train set accuracy and tightness in Figure 5 (left). We observe that increasing capacity via width instead of depth yields a monotone although diminishing increase in accuracy as tightness decreases gradually. The different trends for width and depth agree well with our theoretical results, predicting

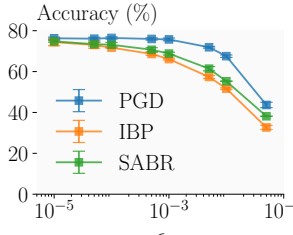 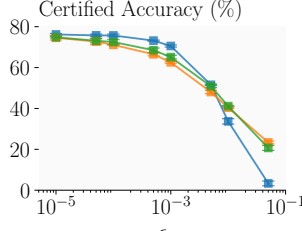 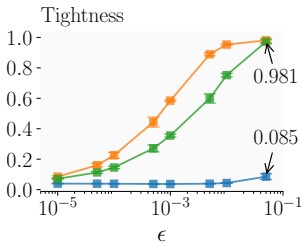

Figure 7: Tightness, standard, and certified accuracy for CNN3 on CIFAR-10, depending on training method and perturbation magnitude $\epsilon$ used for training and evaluation.

that sufficient network width is essential for trained networks (see Theorem 3.10). It can further be explained by the observation that increasing depth, at initialization, reduces tightness exponentially, while increasing width only reduces it polynomially. Intuitively, this suggests that less regularization is required to offset the tightness penalty of increasing network width rather than depth.

As these experiments indicate that optimal architectures for IBP-based training have only moderate depth but large width, we train wider versions of the popular CNN7 using IBP, SABR, and IBP-R, showing results in Table 1 and Figure 6. We observe that this width increase improves certified accuracy in all settings. We note that, while these improvements might seem marginal, they are of similar magnitude as multiple years of progress on certified training methods, see Figure 6 where CROWN-IBP (Zhang et al., 2020) and MTL-IBP (Palma et al., 2023) (the previous SOTA on MNIST) are shown for reference.

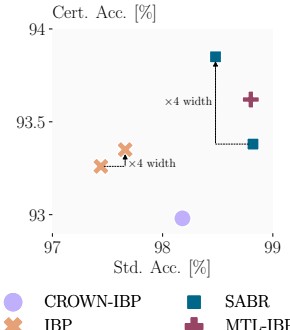

Figure 6: Effect of a 4-fold width increase on certified and standard accuracy for MNIST at $\epsilon = 0.3$.

## 4.2 CERTIFIED TRAINING INCREASES TIGHTNESS

To assess how different training methods affect tightness, we train a CNN3 on CIFAR-10 for a wide range of perturbation magnitudes ($\epsilon \in [10^{-5}, 5 \cdot 10^{-2}]$) using IBP, PGD, and SABR training and illustrate the resulting tightness and accuracies in Figure 7. Recall, that while IBP computes and optimizes a sound over-approximation of the worst-case loss over the whole input region, SABR propagates only a small subregion with IBP, thus yielding an unsound but generally more precise approximation of the worst-case loss. PGD, in contrast, does not use IBP at all but rather trains with samples that approximately maximize the worst-case loss. We observe that training with either IBP-based method increases tightness with perturbation magnitude until networks become almost propagation invariant with $\tau = 0.98$ (see Figure 7, right). This confirms our theoretical results, showing that IBP training increases tightness with $\epsilon$ (see Theorem 3.9). In contrast, training with PGD barely influences tightness. Further, the regularization required for such high tightness comes at the cost of standard accuracies being severely reduced (see Figure 7, left). However, while this reduced standard accuracy translates to smaller certified accuracies for very small perturbation magnitudes ($\epsilon \leq 5 \cdot 10^{-3}$), the increased tightness improves certifiability sufficiently to yield higher certified accuracies for larger perturbation magnitudes ($\epsilon \geq 10^{-2}$).

We further investigate this dependency between (certified) robustness and tightness by varying the subselection ratio $\lambda$ when training with SABR. Recall that $\lambda$ controls the size of the propagated regions for a fixed perturbation magnitude $\epsilon$, recovering IBP for $\lambda = 1$ and PGD for $\lambda = 0$. Plotting results in Figure 8, we observe that while decreasing $\lambda$, severely reduces tightness and thus regularization, it not only leads to increasing natural but also certified accuracies until tightness falls below $\tau < 0.5$ at $\lambda = 0.4$. We observe similar trends when varying the regularization level for other unsound certified training methods, discussed in App. D.1. In Figure 9, we vary the perturbation size $\epsilon$ for three different $\lambda$ and show tightness over the size of the propagation region $\xi = \lambda\epsilon$ for a CNN3 and CIFAR-10. Here, we observe that tightness is dominated by the size of the propagation region $\xi$ and not the robustness specification $\epsilon$, indicating that while training with IBP-bounds increases tightness, the resulting high levels of tightness and thus regularization are not generally necessary for robustness. This helps to explain SABR's success and highlights the potential for developing novel certified training methods that reduce tightness while maintaining sufficient certifiability.

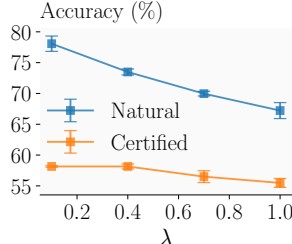 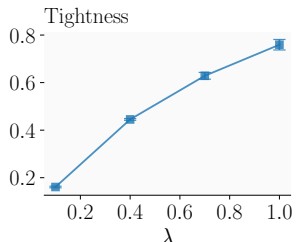 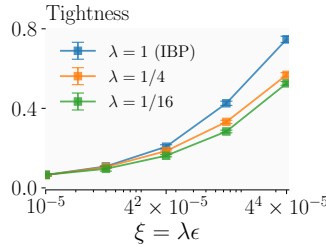

Figure 8: Accuracies and tightness of a CNN7 for CIFAR-10 $\epsilon = \frac{2}{255}$ depending on regularization strength with SABR.

Figure 9: Tightness over propagation region size $\xi$ for SABR.

To study how certified training methods that do not use IBP-bounds at all (COLT) or only as a regularizer with very small weight (IBP-R) affect tightness, we compare tightness, certified, and standard accuracies on a 4-layer CNN (used by COLT and IBP-R) in Table 2. We observe that the orderings of tightness and accuracy are exactly inverted, highlighting the accuracy penalty of a strong regularization for tightness. While both COLT and IBP-R affect a much smaller increase in tightness than SABR or IBP, they still yield networks an order of magnitude tighter than PGD, suggesting that slightly increased tightness might be desirable for certified robustness. This is further corroborated by the more heavily regularizing SABR outperforming IBP-R at larger $\epsilon$ while being outperformed at smaller $\epsilon$.

Table 2: Tightness and accuracies for various training methods on CIFAR-10.

| Method | $\epsilon$ | Accuracy | Tightness | Certified |
|---|---|---|---|---|
| PGD | 2/255 | 81.2 | 0.001 | - |
| | 8/255 | 69.3 | 0.007 | - |
| COLT | 2/255 | 78.4[*] | 0.009 | 60.7[*] |
| | 8/255 | 51.7[*] | 0.057 | 26.7[*] |
| IBP-R | 2/255 | 78.2[*] | 0.033 | 62.0[*] |
| | 8/255 | 51.4[*] | 0.124 | 27.9[*] |
| SABR | 2/255 | 75.6 | 0.182 | 57.7 |
| | 8/255 | 48.2 | 0.950 | 31.2 |
| IBP | 2/255 | 63.0 | 0.803 | 51.3 |
| | 8/255 | 42.2 | 0.977 | 31.0 |

[*] Literature result.

## 5 RELATED WORK

Baader et al. (2020) show that continuous functions can be approximated by IBP-certifiable ReLU networks up to arbitrary precision. Wang et al. (2022b) extend this result to more activation functions and characterize the hardness of such a construction. Wang et al. (2022a) find that IBP-training converges to a global optimum with high probability for sufficient width. Mirman et al. (2022) show that functions with points of non-invertibility can not be precisely approximated with IBP. Zhu et al. (2022) show that width is advantageous while depth is not for approximate average case robustness.

Shi et al. (2021) define *tightness* as the size of the layerwise $\text{Box}^\dagger$, i.e., $\Delta = \overline{z}^\dagger - \underline{z}^\dagger$, rather than its ratio $\tau$ to the size the optimal box (Definition 3.6). They thus study the size of the approximation irrespective of the size of the ground truth, while we study the quality of the approximation. This leads to significantly different insights, e.g., propagation tightness $\tau$ remains the same under scaling of the network weights, while the abstraction size $\Delta$ is scaled proportionally.

Wu et al. (2021) study the relation between empirical adversarial robustness and network width. They observe that in this setting, increased width actually hurts perturbation stability and thus potentially empirical robustness while improving natural accuracy. In contrast, we have shown theoretically and empirically that width is beneficial for certified robustness when training with IBP-based methods.

## 6 CONCLUSION

Motivated by the recent and surprising dominance of IBP-based certified training methods, we investigate its underlying mechanisms and trade-offs. By quantifying the relationship between IBP and optimal BOX bounds with our novel propagation tightness metric, we are able to predict the influence of architecture choices on deep linear networks at initialization and after training. We experimentally confirm the applicability of these results to ReLU networks and show that wider networks improve the performance of state-of-the-art methods, while deeper networks do not. Finally, we show that IBP-based training methods increase propagation tightness, depending on the size of the propagated region, at the cost of strong regularization. This observation not only helps explain the success of recent certified training methods but, in combination with the novel metric of propagation tightness, might constitute a key step towards developing novel training methods, balancing certifiability and the (over-)regularization resulting from propagation tightness.

## REPRODUCIBILITY STATEMENT

We publish our code, trained models, and detailed instructions on how to reproduce our results at `https://github.com/eth-sri/ibp-propagation-tightness`. Additionally, we provide detailed descriptions of all hyper-parameter choices, data sets, and preprocessing steps in App. C.

## ACKNOWLEDGEMENTS

We would like to thank our anonymous reviewers for their constructive comments and insightful questions.

This work has been done as part of the EU grant ELSA (European Lighthouse on Secure and Safe AI, grant agreement no. 101070617) and the SERI grant SAFEAI (Certified Safe, Fair and Robust Artificial Intelligence, contract no. MB22.00088). Views and opinions expressed are however those of the authors only and do not necessarily reflect those of the European Union or European Commission. Neither the European Union nor the European Commission can be held responsible for them.

The work has received funding from the Swiss State Secretariat for Education, Research and Innovation (SERI).

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

## A  ADDITIONAL THEORETICAL RESULTS

Below we present a corollary, formalizing the intutions we provided in Section 3.2.

**Corollary A.1.** *Assume all elements of $\boldsymbol{W}^{(1)}$, $\boldsymbol{W}^{(2)}$ and $\boldsymbol{W}^{(2)}\boldsymbol{W}^{(1)}$ are non-zero and $\boldsymbol{W}^{(2)}\boldsymbol{W}^{(1)}$ is propagation invariant. Then choosing the signs of one row and one column of $\boldsymbol{W}^{(2)}\boldsymbol{W}^{(1)}$ fixes all signs of $\boldsymbol{W}^{(2)}\boldsymbol{W}^{(1)}$.*

*Proof.* For notational reasons, we define $W := W^{(2)}W^{(1)}$. Without loss of generality, assume we know the signs of the first row and the first column, *i.e.*, $W_{1,\cdot}$ and $W_{\cdot,1}$. We prove via a construction of the signs of all elements. The construction is given by the following: whenever $\exists i, j$, such that we know the sign of $W_{i,j}$, $W_{i,j+1}$ and $W_{i+1,j}$, we fix the sign of $W_{i+1,j+1}$ to be positive if there are an odd number of positive elements among $W_{i,j}$, $W_{i,j+1}$ and $W_{i+1,j}$, otherwise negative.

By Theorem 3.4, propagation invariance requires us to fix the sign of the last element in the $W_{i:i+1,j:j+1}$ block in this way. We only need to prove that when this process terminates, we fix the signs of all elements. We show this via recursion.

When $i = 1$ and $j = 1$, we have known the signs of $W_{i,j}$, $W_{i,j+1}$ and $W_{i+1,j}$, thus the sign of $W_{i+1,j+1}$ is fixed. Continuing towards the right, we gradually fix the sign of $W_{2,j+1}$ for $j = 1, \ldots, d-1$. Continuing downwards, we gradually fix the sign of $W_{i+1,2}$ for $i = 1, \ldots, d-1$. Therefore, all signs of the elements of the second row and the second column are fixed. By recursion, we would finally fix all the rows and the columns, thus the whole matrix. $\square$

We extend our result in Section 3.3 to two-layer ReLU networks. The intuition is that when the input data is symmetric around zero, ReLU status (activated or not) is independent to weights and the probability of activation is exactly 0.5.

**Corollary A.2.** *Assume the input distribution is symmetric around zero, i.e., $p_X(x) = p_X(-x)$ for all $x > 0$, and $P(X = 0) = 0$. Then for a two-layer ReLU network $\boldsymbol{f} = \boldsymbol{W}^{(2)} \text{ReLU}(\boldsymbol{W}^{(1)}\boldsymbol{x})$ initialized with i.i.d. Gaussian, the expected local tightness $\tau' \sim \sqrt{2}\tau$, where $\tau$ is the expected tightness of corresponding deep linear network.*

*Proof.* Since the input $X$ is symmetric around 0, the distribution of $\boldsymbol{W}^{(1)}\boldsymbol{x}$ is symmetric around 0 as well, regardless of the initialized weights. By assumption on the input and weight distribution, $P(\boldsymbol{W}^{(1)}\boldsymbol{x} = 0) = 0$, thus $P(\text{ReLU}(\boldsymbol{W}^{(1)}\boldsymbol{x}) = 0) = 0.5$. In addition, the status of activation is independent to the initialized weights. Thus, the effect can be viewed as randomly setting rows of $\boldsymbol{W}^{(1)}$ to zero with probability 0.5. Following Equation (8) and Equation (9), we get that the size of $\text{Box}^{\dagger}$ is scaled by 0.5, and the size of $\text{Box}^*$ is scaled by $\mathbb{E}(\sqrt{\chi^2(d_1/2)})/\mathbb{E}(\sqrt{\chi^2(d_1)}) \sim \frac{\sqrt{2}}{2}$. Therefore, $\tau' \sim \sqrt{2}\tau$. $\square$

We perform a Monte-Carlo estimation of the ratio $\tau'/\tau$ with a two-layer fully connected network and a two-layer convolutional network on MNIST. The estimation is $1.4167 \pm 0.0059$ and $1.4228 \pm 0.0368$, respectively, which is close to the theoretical value $\sqrt{2} \approx 1.4142$. This confirms the correctness of our theoretical analysis and its generalization even to convolutional networks which do not fully satisfy the assumption.

## B  DEFERRED PROOFS

**Proof of Lemma 3.1**

Here we prove Lemma 3.1, restated below for convenience.

**Lemma 3.1.** *Any $\mathcal{C}^0$ continuous classifier $\boldsymbol{f}$, computing the logit difference $y_i^{\Delta} := y_i - y_t, \forall i \neq t$, is robustly correct on $\mathcal{B}^{\epsilon}(\boldsymbol{x})$ if and only if $\text{Box}^*(\boldsymbol{f}, \mathcal{B}^{\epsilon}(\boldsymbol{x})) \subseteq \mathbb{R}_{<0}^{c-1}$, i.e., $\bar{y}_i^{\Delta^*} < 0, \forall i \neq t$.*

*Proof.* On the one hand, assume $y_i - y_{\text{true}} < 0$ for all $i$. Then for the $i^{th}$ output dimension, the optimal bounding box is $\max y_i - y_{\text{true}}$. Since the classifier is continuous, $\boldsymbol{f}(\mathcal{B}(\boldsymbol{x}, \boldsymbol{\epsilon}))$ is a closed and

bounded set. Therefore, by extreme value theorem, $\exists \eta \in \mathcal{B}(\boldsymbol{x}, \boldsymbol{\epsilon})$ such that $\eta = \arg\max y_i - y_{\text{true}}$, thus $\max y_i - y_{\text{true}} < 0$. Since this holds for every $i$, $\text{Box}^*(\boldsymbol{f}, \mathcal{B}(\boldsymbol{x}, \boldsymbol{\epsilon})) \subseteq \mathcal{R}_{<0}^{K-1}$.

On the other hand, assume $\text{Box}^*(\boldsymbol{f}, \mathcal{B}(\boldsymbol{x}, \boldsymbol{\epsilon})) \subseteq \mathcal{R}_{<0}^{K-1}$. Since $\boldsymbol{f}(\mathcal{B}(\boldsymbol{x}, \boldsymbol{\epsilon})) \subseteq \text{Box}^*(\boldsymbol{f}, \mathcal{B}(\boldsymbol{x}, \boldsymbol{\epsilon})) \subseteq \mathcal{R}_{<0}^{K-1}$, we get $y_i - y_{\text{true}} < 0$ for all $i$. $\square$

**Proof of Theorem 3.2**

We first prove Theorem 3.2 for a 2-layer DLN as Lemma B.1.

**Lemma B.1.** *For a two-layer DLN* $\boldsymbol{f} = \boldsymbol{W}^{(2)}\boldsymbol{W}^{(1)}$, $(\overline{\boldsymbol{z}}^* - \underline{\boldsymbol{z}}^*)/2 = \left|W^{(2)}W^{(1)}\right| \boldsymbol{\epsilon}$ and $(\overline{\boldsymbol{z}}^\dagger - \underline{\boldsymbol{z}}^\dagger)/2 = \left|W^{(2)}\right|\left|W^{(1)}\right| \boldsymbol{\epsilon}$. *In addition,* $\text{Box}^*$ *and* $\text{Box}^\dagger$ *have the same center* $\boldsymbol{f}(\boldsymbol{x})$.

*Proof.* First, assume $W^{(1)} \in \mathcal{R}^{d_1 \times d_0}$, $W^{(2)} \in \mathcal{R}^{d_2 \times d_1}$ and $B_i = [-1, 1]^{d_i}$ for $i = 0, 1, 2$, where $d_i \in \mathcal{Z}_+$ are some positive integers. The input box can be represented as $\text{diag}(\epsilon_0)B_0 + b$ for $\epsilon_0 = \epsilon$.

For a single linear layer, the box propagation yields

$$\text{Box}(W^{(1)}(\text{diag}(\epsilon_0)B_0 + b)) = \text{Box}(W^{(1)}\text{diag}(\epsilon_0)B_0) + W^{(1)}b$$

$$= \text{diag}\left(\sum_{j=1}^{d_0} |W_{i,j}^{(1)}|\epsilon_0[j]\right) B_1 + W^{(1)}b$$

$$:= \text{diag}(\epsilon_1)B_1 + W^{(1)}b. \tag{5}$$

Applying Equation (5) iteratively, we get the explicit formula of layer-wise propagation for two-layer linear network:

$$\text{Box}(W^{(2)}\text{Box}(W^{(1)}(\text{diag}(\epsilon_0)B_0 + b)))$$

$$= \text{Box}\left(W^{(2)}(\text{diag}(\epsilon_1)B_1 + W^{(1)}b)\right)$$

$$= \text{diag}\left(\sum_{k=1}^{d_1} |W_{i,k}^{(2)}|\epsilon_1[k]\right) B_2 + W^{(2)}W^{(1)}b$$

$$= \text{diag}\left(\sum_{j=1}^{d_0} \epsilon_0[j]\left(\sum_{k=1}^{d_1} |W_{i,k}^{(2)}W_{k,j}^{(1)}|\right)\right) B_2 + W^{(2)}W^{(1)}b. \tag{6}$$

Applying Equation (5) on $W := W^{(2)}W^{(1)}$, we get the explicit formula of the tightest box:

$$\text{Box}(W^{(2)}W^{(1)}(\text{diag}(\epsilon_0)B_0 + b))$$

$$= \text{diag}\left(\sum_{j=1}^{d_0} |(W^{(2)}W^{(1)})_{i,j}|\epsilon_0[j]\right) B_2 + W^{(2)}W^{(1)}b$$

$$= \text{diag}\left(\sum_{j=1}^{d_0} \epsilon_0[j]\left|\sum_{k=1}^{d_1} W_{i,k}^{(2)}W_{k,j}^{(1)}\right|\right) B_2 + W^{(2)}W^{(1)}b. \tag{7}$$

$\square$

Now, we use induction and Lemma B.1 to prove Theorem 3.2, restated below for convenience. The key insight is that a multi-layer DLN is equivalent to a single-layer linear network. Thus, we can group layers together and view general DLNs as two-layer DLNs.

**Theorem 3.2** (Box Propagation). *For an $L$-layer DLN $\boldsymbol{f} = \Pi_{k=1}^{L}\boldsymbol{W}^{(k)}$, we obtain the box centres $\dot{\boldsymbol{z}}^* = \dot{\boldsymbol{z}}^\dagger = \boldsymbol{f}(\boldsymbol{x})$ and the radii*

$$\frac{\overline{\boldsymbol{z}}^* - \underline{\boldsymbol{z}}^*}{2} = \left|\Pi_{k=1}^{L}\boldsymbol{W}^{(k)}\right| \boldsymbol{\epsilon}, \quad \text{and} \quad \frac{\overline{\boldsymbol{z}}^\dagger - \underline{\boldsymbol{z}}^\dagger}{2} = \left(\Pi_{k=1}^{L}\left|\boldsymbol{W}^{(k)}\right|\right) \boldsymbol{\epsilon}. \tag{4}$$

*Proof.* For $L = 2$, by Lemma B.1, the result holds. Assume for $L \le m$, the result holds. Therefore, for $L = m + 1$, we group the first $m$ layers as a single layer, resulting in a "two" layer equivalent network. Thus, $(\overline{z}^* - \underline{z}^*)/2 = \left| W^{(m+1)} \Pi_{k=1}^m W^{(k)} \right| \epsilon = \left| \Pi_{k=1}^L W^{(k)} \right| \epsilon$. Similarly, by Equation (5), we can prove $(\overline{z}^* - \underline{z}^*)/2 = \left( \left| W^{(m+1)} \right| \Pi_{k=1}^m \left| W^{(k)} \right| \right) \epsilon = \left( \Pi_{k=1}^L \left| W^{(k)} \right| \right) \epsilon$. The claim about center follows by induction similarly. $\square$

**Proof of Lemma 3.3**

Here, we prove Lemma 3.3, restated below for convenience.

**Lemma 3.3** (Propagation Invariance). *A two-layer DLN $f = W^{(2)} W^{(1)}$ is propagation invariant if and only if for every fixed $(i, j)$, we have $\left| \sum_k W_{i,k}^{(2)} \cdot W_{k,j}^{(1)} \right| = \sum_k |W_{i,k}^{(2)} \cdot W_{k,j}^{(1)}|$, i.e., either $W_{i,k}^{(2)} \cdot W_{k,j}^{(1)} \ge 0$ for all $k$ or $W_{i,k}^{(2)} \cdot W_{k,j}^{(1)} \le 0$ for all $k$.*

*Proof.* We prove the statement via comparing the box bounds. By Lemma B.1, we need $\left| \sum_{k=1}^{d_1} W_{i,k}^{(2)} W_{k,j}^{(1)} \right| = \sum_{k=1}^{d_1} |W_{i,k}^{(2)} W_{k,j}^{(1)}|$. The triangle inequality of absolute function says this holds if and only if $W_{i,k}^{(2)} W_{k,j}^{(1)} \ge 0$ for all $k$ or $W_{i,k}^{(2)} W_{k,j}^{(1)} \le 0$ for all $k$. $\square$

**Proof of Theorem 3.4**

Here, we prove Theorem 3.4, restated below for convenience.

**Theorem 3.4** (Non-Propagation Invariance). *Assume $\exists i, i', j, j'$, such that $W_{\cdot,j}^{(1)}$, $W_{\cdot,j'}^{(1)}$, $W_{i,\cdot}^{(2)}$ and $W_{i',\cdot}^{(2)}$ are all non-zero. If $(W^{(2)} W^{(1)})_{i,j} \cdot (W^{(2)} W^{(1)})_{i,j'} \cdot (W^{(2)} W^{(1)})_{i',j} \cdot (W^{(2)} W^{(1)})_{i',j'} < 0$, then $f = W^{(2)} W^{(1)}$ is not propagation invariant.*

*Proof.* The assumption $(W^{(2)} W^{(1)})_{i,j} \cdot (W^{(2)} W^{(1)})_{i,j'} \cdot (W^{(2)} W^{(1)})_{i',j} \cdot (W^{(2)} W^{(1)})_{i',j'} < 0$ implies three elements are of the same sign while the other element has a different sign. Without loss of generality, assume $(W^{(2)} W^{(1)})_{i',j'} < 0$ and the rest three are all positive.

Assume $W^{(2)} W^{(1)}$ is propagation invariant. By Lemma 3.3, this means $W_{i,\cdot}^{(2)}.\text{sign} = W_{\cdot,j}^{(1)}.\text{sign}$, $W_{i,\cdot}^{(2)}.\text{sign} = W_{\cdot,j'}^{(1)}.\text{sign}$, $W_{i',\cdot}^{(2)}.\text{sign} = W_{\cdot,j}^{(1)}.\text{sign}$ and $W_{i',\cdot}^{(2)}.\text{sign} = -W_{\cdot,j'}^{(1)}.\text{sign}$. Therefore, we have $-W_{\cdot,j'}^{(1)}.\text{sign} = W_{\cdot,j'}^{(1)}.\text{sign}$, which implies all elements of $W_{\cdot,j'}^{(1)}$ must be zero. However, this results in $(W^{(2)} W^{(1)})_{i,j'} = 0$, a contradiction. $\square$

**Proof of Lemma 3.7**

Here, we prove Lemma 3.7, restated below for convenience.

**Lemma 3.7** (Initialization Tightness w.r.t. Width). *Given a 2-layer DLN with weight matrices $W^{(1)} \in \mathcal{R}^{d_1 \times d_0}$, $W^{(2)} \in \mathcal{R}^{d_2 \times d_1}$ with i.i.d. entries from $\mathcal{N}(0, \sigma_1^2)$ and $\mathcal{N}(0, \sigma_2^2)$ (together denoted as $\theta$), we obtain the expected tightness $\tau(d_1) = \frac{\mathbb{E}_\theta(\underline{z}^* - \overline{z}^*)}{\mathbb{E}_\theta(\overline{z}^\dagger - \underline{z}^\dagger)} = \frac{\sqrt{\pi} \Gamma(\frac{1}{2}(d_1+1))}{d_1 \Gamma(\frac{1}{2} d_1)} \in \Theta(\frac{1}{\sqrt{d_1}})$.*

*Proof.* We first compute the size of the layer-wisely propagated box. From Equation (6), we get that for the $i$-th dimension,

$$\mathbb{E}(u_i - l_i) = \mathbb{E}\left( \sum_{j=1}^{d_0} \epsilon_0[j] \left( \sum_{k=1}^{d_1} |W_{i,k}^{(2)} W_{k,j}^{(1)}| \right) \right)$$

$$= \sum_{j=1}^{d_0} \epsilon_0[j] \left( \sum_{k=1}^{d_1} \mathbb{E}(|W_{i,k}^{(2)}|) \cdot \mathbb{E}(|W_{k,j}^{(1)}|) \right)$$

$$= \sigma_1 \sigma_2 \sum_{j=1}^{d_0} \epsilon_0[j] \left( \sum_{k=1}^{d_1} \mathbb{E}(|\mathcal{N}(0,1)|)^2 \right).$$

Since $\mathbb{E}(|\mathcal{N}(0,1)|) = \sqrt{\frac{2}{\pi}}$ [1], we have

$$\mathbb{E}(u_i - l_i) = \frac{2}{\pi}\sigma_1\sigma_2 d_1 \|\epsilon_0\|_1. \tag{8}$$

Now we compute the size of the tightest box. From Equation (7), we get that for the $i$-th dimension,

$$\mathbb{E}(u_i^* - l_i^*) = \mathbb{E}\left(\sum_{j=1}^{d_0} \epsilon_0[j] \left|\sum_{k=1}^{d_1} W_{i,k}^{(2)} W_{k,j}^{(1)}\right|\right) = \sigma_1\sigma_2 \sum_{j=1}^{d_0} \epsilon_0[j] \mathbb{E}\left(\left|\sum_{k=1}^{d_1} X_k Y_k\right|\right),$$

where $X_k$ and $Y_k$ are i.i.d. standard Gaussian random variables. Using the law of total expectation, we have

$$\mathbb{E}\left(\left|\sum_{k=1}^{d_1} X_k Y_k\right|\right) = \mathbb{E}\left(\mathbb{E}\left(\left|\sum_{k=1}^{d_1} X_k Y_k\right| \,\middle|\, Y_k\right)\right)$$

$$= \mathbb{E}\left(\mathbb{E}\left(\left|\mathcal{N}(0, \sum_{k=1}^{d_1} Y_k^2)\right| \,\middle|\, Y_k\right)\right)$$

$$= \sqrt{\frac{2}{\pi}}\mathbb{E}\left(\sqrt{\sum_{k=1}^{d_1} Y_k^2}\right)$$

$$= \sqrt{\frac{2}{\pi}}\mathbb{E}(\sqrt{\chi^2(d_1)}).$$

Since $\mathbb{E}(\sqrt{\chi^2(d_1)}) = \sqrt{2}\Gamma(\frac{1}{2}(d_1+1))/\Gamma(\frac{1}{2}d_1)$, [2] we have

$$\mathbb{E}(u_i^* - l_i^*) = \frac{2}{\sqrt{\pi}}\sigma_1\sigma_2 \|\epsilon_0\|_1 \Gamma(\frac{1}{2}(d_1+1))/\Gamma(\frac{1}{2}d_1). \tag{9}$$

Combining Equation (8) and Equation (9), we have:

$$\frac{\mathbb{E}(u_i - l_i)}{\mathbb{E}(u_i^* - l_i^*)} = \frac{d_1 \Gamma(\frac{1}{2}d_1)}{\sqrt{\pi}\Gamma(\frac{1}{2}(d_1+1))}. \tag{10}$$

To see the asymptotic behavior, use $\Gamma(x+\alpha)/\Gamma(x) \sim x^\alpha$, [3] we have

$$\frac{\mathbb{E}(u_i - l_i)}{\mathbb{E}(u_i^* - l_i^*)} \sim \frac{1}{\sqrt{\pi}}d_1^{\frac{1}{2}}. \tag{11}$$

To establish the bounds on the minimum expected slackness, we use Lemma B.2. $\qquad \square$

**Lemma B.2.** *Let* $g(n) := \frac{n\Gamma(\frac{1}{2}n)}{\sqrt{\pi}\Gamma(\frac{1}{2}(n+1))}$. $g(n)$ *is monotonically increasing for* $n \geq 1$. *Thus, for* $n \geq 2$, $g(n) \geq g(2) > 1.27$.

*Proof.* Using polygamma function $\psi^{(0)}(z) = \Gamma'(z)/\Gamma(z)$, [4] we have

$$g'(n) \propto 1 + \frac{1}{2}n\left(\psi^{(0)}\left(\frac{1}{2}n\right) - \psi^{(0)}\left(\frac{1}{2}(n+1)\right)\right).$$

---

[1] https://en.wikipedia.org/wiki/Half-normal_distribution
[2] https://en.wikipedia.org/wiki/Chi_distribution
[3] https://en.wikipedia.org/wiki/Gamma_function#Stirling's_formula
[4] https://en.wikipedia.org/wiki/Polygamma_function

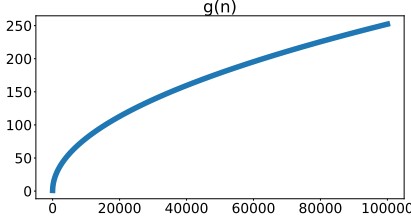 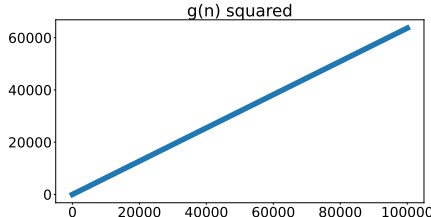

Figure 10: $g(n)$ and $g^2(n)$ visualized.

Using the fact that $\psi^{(0)}(z)$ is monotonically increasing for $z > 0$ and $\psi^{(0)}(z+1) = \psi^{(0)}(z) + \frac{1}{z}$, we have

$$1 + \frac{1}{2}n \left( \psi^{(0)}\left( \frac{1}{2}n \right) - \psi^{(0)}\left( \frac{1}{2}(n+1) \right) \right)$$
$$> 1 + \frac{1}{2}n \left( \psi^{(0)}\left( \frac{1}{2}n \right) - \psi^{(0)}\left( \frac{1}{2}n + 1 \right) \right)$$
$$= 1 + \frac{1}{2}n \left( -\frac{2}{n} \right)$$
$$= 0.$$

Therefore, $g'(n)$ is strictly positive for $n \geq 1$, and thus $g(n)$ is monotonically increasing for $n \geq 1$. □

As a final comment, we visualize $g(n)$ in Figure 10. As expected, $g(n)$ is monotonically increasing in the order of $O(\sqrt{n})$.

**Proof of Corollary 3.8**

Here, we prove Corollary 3.8, restated below for convenience.

**Corollary 3.8** (Initialization Tightness w.r.t. Depth). *The expected tightness of an $L$-layer DLN $\boldsymbol{f}$ with minimum internal dimension $d_{min}$ is at most $\tau \leq \tau(d_{min})^{\lfloor \frac{L}{2} \rfloor}$ at initialization.*

*Proof.* This is pretty straightforward and only requires a coarse application of Lemma 3.7. Without loss of generality, we assume $L$ is even. If $L$ is odd, then we simply discard the slackness introduced by the last layer, *i.e.*, assume the last layer does not introduce additional slackness.

We group the $2i-1$-th and $2i$-th layer as a new layer. By Lemma 3.7, these $L/2$ subnetworks all introduce an additional slackness factor of $\tau$. Note that Equation (8) implies that the size of the output box is proportional to the size of the input box. Therefore, the layer-wisely propagated box of $\Pi_{i=1}^{L} W_i$ is $\tau^{L/2}$ looser than the layer-wisely propagated box of $\Pi_{j=1}^{L/2}(W_{2j-1}W_{2j})$. In addition, the size of the tightest box for $\Pi_{i=1}^{L} W_i$ is upper bounded by layer-wisely propagating $\Pi_{j=1}^{L/2}(W_{2j-1}W_{2j})$. Therefore, the minimum expected slackness is lower bounded by $\tau^{L/2}$. □

**Proof of Theorem 3.9**

Here, we prove Theorem 3.9, restated below for convenience.

**Theorem 3.9** (IBP Training Increases Tightness). *Assume homogenous tightness,* i.e., $\boldsymbol{W}^* = \tau \boldsymbol{W}^\dagger$, *and* $\frac{\|\nabla_\theta \boldsymbol{W}_{ij}^*\|_2}{\boldsymbol{W}_{ij}^*} \leq \frac{1}{2} \frac{\|\nabla_\theta \boldsymbol{W}_{ij}^\dagger\|_2}{\boldsymbol{W}_{ij}^\dagger}$ *for all* $i, j$, *then, the gradient difference between the IBP and standard loss is aligned with an increase in tightness,* i.e., $\langle \nabla_\theta(R(\boldsymbol{\epsilon}) - R(0)), \nabla_\theta \tau \rangle \leq 0$ *for all* $\boldsymbol{\epsilon} > 0$.

*Proof.* We prove a stronger claim: $\langle \nabla_\theta(R(\boldsymbol{\epsilon} + \Delta\boldsymbol{\epsilon}) - R(\boldsymbol{\epsilon})), \nabla_\theta \tau \rangle \leq 0$ for all $\boldsymbol{\epsilon} \geq 0$ and $\Delta\boldsymbol{\epsilon} > 0$. Let $\boldsymbol{\epsilon} = \boldsymbol{0}$ yields the theorem.

We prove the claim for $\Delta\epsilon \to 0$. For large $\Delta\epsilon$, we can break it into $R(\epsilon + \Delta\epsilon) - R(\epsilon) = \sum_{i=1}^{n} R(\epsilon + \frac{i}{n}\Delta\epsilon) - R(\epsilon + \frac{i-1}{n}\Delta\epsilon)$, thus proving the claim since each summand satisfies the theorem.

Let $L_1 = R(\epsilon)$ and $L_2 = R(\epsilon + \Delta\epsilon)$. By Taylor expansion, we have $L_2 = L_1 + \Delta\epsilon^\top W^\dagger \nabla_{\boldsymbol{u}} g = L_1 + \frac{1}{\tau}\Delta\epsilon^\top W^* \nabla_{\boldsymbol{u}} g$, where $\nabla_{\boldsymbol{u}} g = \nabla_{\boldsymbol{u}} g(\boldsymbol{u})$ evaluated at $\boldsymbol{u} = W^\dagger \epsilon$. Note that the increase of $\epsilon$ would increase the risk, thus $\nabla_{\boldsymbol{u}} g \geq \mathbf{0}$.

For the $i^{\text{th}}$ parameter $\theta_i$, $\nabla_{\theta_i}(L_2 - L_1)\nabla_{\theta_i}\tau = \frac{1}{\tau^2}\Delta\epsilon^\top(\tau\nabla_{\theta_i} W^* - W^*\nabla_{\theta_i}\tau)\nabla_{\boldsymbol{u}} g \nabla_{\theta_i}\tau$. Thus, $\langle\nabla_\theta(L_2 - L_1), \nabla_\theta\tau\rangle = \frac{1}{\tau^2}\Delta\epsilon^\top(\tau\sum_i \nabla_{\theta_i}\tau \cdot \nabla_{\theta_i} W^* - W^*\|\nabla_\theta\tau\|_2^2)\nabla_{\boldsymbol{u}} g$. Since $\Delta\epsilon > \mathbf{0}$ and $\nabla_{\boldsymbol{u}} g \geq \mathbf{0}$, it suffices to prove that $\tau\sum_i \nabla_{\theta_i}\tau \cdot \nabla_{\theta_i} W^* - W^*\|\nabla_\theta\tau\|_2^2$ is nonpositive, $i.e.$, $\tau\langle\nabla_\theta\tau, \nabla_\theta W_{ij}^*\rangle - W_{ij}^*\|\nabla_\theta\tau\|_2^2$ is nonpositive for every $i, j$.

Since $\|\boldsymbol{u}\|_2\|\boldsymbol{v}\|_2 \geq \langle\boldsymbol{u}, \boldsymbol{v}\rangle$, we have

$$\frac{\|\nabla_\theta W_{ij}^*\|_2}{W_{ij}^*} \leq \frac{1}{2}\frac{\|\nabla_\theta W_{ij}^\dagger\|_2}{W_{ij}^\dagger}$$

$$\Rightarrow \|\nabla_\theta \log W^\dagger\|_2 \geq 2\|\nabla_\theta \log W^*\|_2$$

$$\Rightarrow \|\nabla_\theta \log W^\dagger\|_2^2 \geq 2\langle\nabla_\theta \log W^\dagger, \nabla_\theta \log W^*\rangle$$

Therfore, $\|\nabla_\theta \log \tau\|_2^2 = \|\nabla_\theta(\log W_{ij}^* - \log W_{ij}^\dagger)\|_2^2 = \|\nabla_\theta \log W_{ij}^*\|_2^2 - 2\langle\nabla_\theta \log W^\dagger, \nabla_\theta \log W^*\rangle + \|\nabla_\theta \log W^\dagger\|_2^2 \geq \|\nabla_\theta \log W_{ij}^*\|_2^2$. This means $\frac{\|\nabla_\theta W_{ij}^*\|_2}{W_{ij}^*} \leq \frac{\|\nabla_\theta\tau\|_2}{\tau}$, thus $W_{ij}^*\|\nabla_\theta\tau\|_2^2 \geq \tau\|\nabla_\theta\tau\|_2\|\nabla_\theta W_{ij}^*\|_2 \geq \tau\langle\nabla_\theta\tau, \nabla_\theta W_{ij}^*\rangle$, which fulfills our goal. $\square$

**Proof of Theorem 3.10**

Here, we prove Theorem 3.10, restated below for convenience.

**Theorem 3.10** (Box Reconstruction Error). *Consider the linear embedding and reconstruction $\hat{\boldsymbol{x}} = U_k U_k^\top \boldsymbol{x}$ of a $d$ dimensional data distribution $\boldsymbol{x} \sim \mathcal{X}$ into a $k$ dimensional space with $d \gg k$ and eigenmatrices $U$ drawn uniformly at random from the orthogonal group. Propagating the input box $\mathcal{B}^\epsilon(\boldsymbol{x})$ layer-wise and optimally, thus, yields $\mathcal{B}^{\boldsymbol{\delta}^\dagger}(\hat{\boldsymbol{x}})$, and $\mathcal{B}^{\boldsymbol{\delta}^*}(\hat{\boldsymbol{x}})$, respectively. Then, we have, (i) $\mathbb{E}(\delta_i/\epsilon) = ck \in \Theta(k)$ for a positive constant $c$ depending solely on $d$ and $c \to \frac{2}{\pi} \approx 0.64$ for large $d$; and (ii) $\mathbb{E}(\delta_i^*/\epsilon) \to \frac{2}{\sqrt{\pi}}\frac{\Gamma(\frac{1}{2}(k+))}{\Gamma(\frac{1}{2}k)} \in \Theta(\sqrt{k})$.*

*Proof.* Since box propagation for linear functions maps the center of the input box to the center of the output box, the center of the output box is exactly $\hat{X}$. By Lemma B.1, we have $\boldsymbol{\delta} = |U_k||U_k|^\top \epsilon \mathbf{1}$. For notational simplicity, let $V = |U_k|$, thus

$$\boldsymbol{\delta}_i = \sum_{j=1}^{k} V_{ij}(\sum_{p=1}^{d} V_{jp}^\top \epsilon) = \epsilon\sum_{p=1}^{d}\sum_{j=1}^{k} V_{ij}V_{pj} = \epsilon\sum_{j=1}^{k} V_{ij}\|V_{:j}\|_1.$$

Therefore, $\mathbb{E}\boldsymbol{\delta}_i/\epsilon = \sum_{j=1}^{k}\mathbb{E}(V_{ij}\|V_{:j}\|_1) = ck$, where $c = \mathbb{E}(V_{ij}\|V_{:j}\|_1)$. Since $V_{:j}$ is the absolute value of a column of the orthogonal matrix uniformly drawn, $V_{:j}$ itself is the absolute value of a vector drawn uniformly from the unit hyper-ball. By Cook (1957) and Marsaglia (1972), $V_{:j}$ is equivalent in distribution to *i.i.d.* draw samples from the standard Gaussian for each dimension and then normalize it by its $L_2$ norm. For notational simplicity, let $V_{:j} \overset{d}{=} v = |u|$, where $u = \hat{u}/\|\hat{u}\|_2$ and all dimensions of $\hat{u}$ are *i.i.d.* drawn from the standard Gaussian distribution, thus $c = \mathbb{E}(v_1\|v\|_1)$.

Expanding $\|v\|_1$, we have $c = \mathbb{E}(v_1^2) + \sum_{i=2}^{d}\mathbb{E}(v_1 v_i) = \frac{1}{d}\mathbb{E}(\|v\|_2^2) + (d-1)\mathbb{E}(v_1 v_2) = \frac{1}{d} + (d-1)\mathbb{E}(v_1 v_2)$. From page 20 of Pinelis & Molzon (2016), we know each entry in $u$ converges to $\mathcal{N}(0, 1/d)$ at $O(1/d)$ speed in Kolmogorov distance. In addition, $\mathbb{E}(v_1 v_2) = \mathbb{E}(\mathbb{E}(v_2 \mid v_1) \cdot v_1) = \mathbb{E}(v_1\sqrt{1 - v_1^2})\mathbb{E}(v_2')$, where $v'$ is the absolute value of a random vector uniformly drawn from the $d - 1$ dimensional sphere. Therefore, for large $d$, $c = (d-1)\mathbb{E}(v_1\sqrt{1 - v_1^2})\mathbb{E}(v_2') = (d-1)\mathbb{E}(v_1)\mathbb{E}(v_2') = (d-1)\mathbb{E}(|\mathcal{N}(0, 1/d)|)\mathbb{E}(|\mathcal{N}(0, 1/(d-1))|) = \frac{2}{\pi}$.

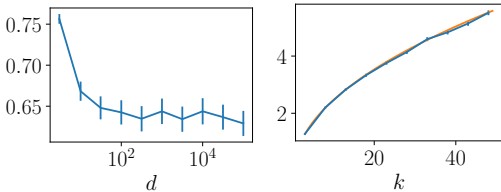

Figure 11: Monte-Carlo estimations of Theorem 3.10. Result bases on 10000 samples for each $d$. Left: $c$ plotted against $d$ in log scale. Right: $\mathbb{E}(\delta_i^*)$ plotted against $k$ for $d = 2000$ (blue), together with the theoretical predictions (orange).

To show how good the asymptotic result is, we run Monte-Carlo to get the estimation of $c$. As shown in the left of Figure 11, the Monte-Carlo result is consistent to this theorem. In addition, it converges very quickly, *e.g.*, stablizing at 0.64 when $d \geq 100$.

Now we start proving (2). By Lemma B.1, we have $\delta^* = |U_k U_k^\top| \epsilon \mathbf{1}$. Thus,

$$\mathbb{E}(\delta_i^*/\epsilon) = \sum_{j=1}^d \mathbb{E}\left|\sum_{p=1}^k U_{ip}U_{jp}\right| = \sum_{j\neq i} \mathbb{E}\left|\sum_{p=1}^k U_{ip}U_{jp}\right| + \mathbb{E}(\sum_{p=1}^k U_{ip}^2) = (d-1)\mathbb{E}\left|\sum_{p=1}^k U_{ip}U_{jp}\right| + \frac{k}{d}.$$

In addition, we have

$$(d-1)\mathbb{E}\left|\sum_{p=1}^k U_{ip}U_{jp}\right| = (d-1)\mathbb{E}_{U_i}\left(\mathbb{E}_{U_j}\left(\left|\sum_{p=1}^k U_{ip}U_{jp}\right|\,\Big|\,U_i\right)\right)$$

$$\to (d-1)\mathbb{E}_{U_i}\left(\mathbb{E}\left|\mathcal{N}\left(0, \frac{\sum_{p=1}^k U_{ip}^2}{d-1}\right)\right|\right)$$

$$= (d-1)\sqrt{\frac{2}{\pi(d-1)}}\mathbb{E}\sqrt{\sum_{p=1}^k U_{ip}^2}$$

$$= \sqrt{\frac{2(d-1)}{\pi}}\mathbb{E}\sqrt{\frac{1}{d}\chi^2(k)}$$

$$\to \frac{2}{\sqrt{\pi}}\frac{\Gamma(\frac{1}{2}(k+1))}{\Gamma(\frac{1}{2}k)},$$

where we use again that for large $d$, the entries of a column tends to Gaussian. This proves (2). The expected tightness follows by definition, *i.e.*, dividing the result of (1) and (2). □

The right of Figure 11 plots the Monte-Carlo estimations against our theoretical results. Clearly, this confirms our result.

## C  EXPERIMENTAL DETAILS

### C.1  DATASET

We use the MNIST (LeCun et al., 2010) and CIFAR-10 (Krizhevsky et al., 2009) datasets for our experiments. Both are open-source and freely available. For MNIST, we do not apply any preprocessing or data augmentation. For CIFAR-10, we normalize images with their mean and standard deviation and, during training, first apply 2-pixel zero padding and then random cropping to $32 \times 32$.

### C.2  MODEL ARCHITECTURE

We follow previous works (Shi et al., 2021; Müller et al., 2022b; Mao et al., 2023) and use a 7-layer convolutional network `CNN7` in most experiments. We also use a simplified 3-layer convolutional network `CNN3` in Section 4.2. Details about them can be found in the released code.

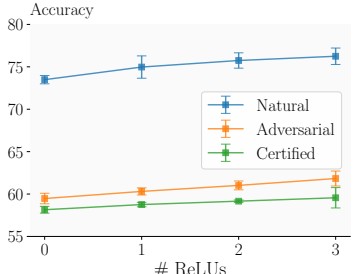 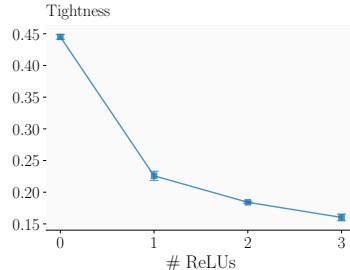

Figure 12: Accuracies and tightness of a `CNN7` for CIFAR-10 $\epsilon = \frac{2}{255}$ depending on regularization strength with STAPS.

### C.3 TRAINING

Following previous works (Müller et al., 2022b; Mao et al., 2023), we use the initialization, warm-up regularization, and learning schedules introduced by Shi et al. (2021). Specifically, for MNIST, the first 20 epochs are used for $\epsilon$-scheduling, increasing $\epsilon$ smoothly from 0 to the target value. Then, we train an additional 50 epochs with two learning rate decays of 0.2 at epochs 50 and 60, respectively. For CIFAR-10, we use 80 epochs for $\epsilon$-annealing, after training models with standard training for 1 epoch. We continue training for 80 further epochs with two learning rate decays of 0.2 at epochs 120 and 140, respectively. The initial learning rate is $5 \times 10^{-3}$ and the gradients are clipped to an $L_2$ norm of at most 10.0 before every step.

### C.4 CERTIFICATION

We apply MN-BAB (Ferrari et al., 2022), a sate-of-the-art (Brix et al., 2023; Müller et al., 2022a) verifier based on multi-neuron constraints (Müller et al., 2022c; Singh et al., 2019a) and the branch-and-bound paradigm (Bunel et al., 2020) to certify all models. MN-BAB is a state-of-the-art complete certification method built on multi-neuron relaxations. For Table 1, we use the same hyperparameters for MN-BAB as Müller et al. (2022b) and set the timeout to 1000 seconds. For other experiments, we use the same hyperparameters but reduce timeout to 200 seconds for efficiency reasons.

## D EXTENDED EMPIRICAL EVALUATION

### D.1 STAPS-TRAINING AND REGULARIZATION LEVEL

To confirm our observations on the interaction of regularization level, accuracies, and propagation tightness from Section 4.2, we extend our experiments to STAPS (Mao et al., 2023), an additional state-of-the-art certified training method beyond SABR (Müller et al., 2022b). Recall that STAPS combines SABR with adversarial training as follows. The model is first (conceptually) split into a feature extractor and classifier. Then, during training IBP is used to propagate the input region through the feature extractor yielding box bounds in the model's latent space. Then, adversarial training with PGD is conducted over the classifier using these box bounds as input region. As IBP leads to an over-approximation while PGD leads to an under-approximation, STAPS induces more regularization as fewer (ReLU) layers are included in the classifier.

We visualize the result of thus varying regularization levels by changing the number of ReLU layers in the classifier in Figure 12. We observe very similar trends as for SABR in Figure 8, although to a lesser extent, as 0 ReLU layers in the classifier still recovers SABR and not standard IBP. Again, decreasing regularization (increasing the number of ReLU layers in the classifier) leads to reducing tightness and increasing standard and certified accuracies.

### D.2 TIGHTNESS AND PROPAGATION REGION SIZE

We repeat the experiment illustrated in Figure 9 for CIFAR-10 on MNIST using a `CNN3` in Figure 13. We again observe the propagation region size $\xi$ dominating the tightness (except for very large perturbation sizes of $\epsilon > 0.2$), and smaller perturbation magnitudes leading to slightly larger tightness.

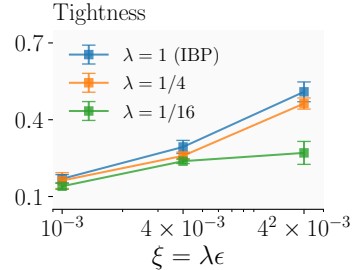

Figure 13: Tightness over propagation region size $\xi$ for SABR and MNIST.

### D.3 TIGHTNESS APPROXIMATION ERROR

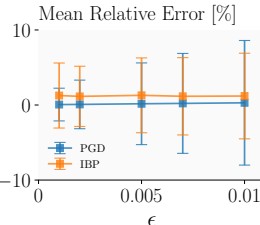

To investigate the approximation quality of our local tightness as defined in Definition 3.6, we compare it against the true tightness computed using MILP (Tjeng et al., 2019). We confirm our results from Figure 2 on MNIST on CIFAR-10 in Figure 14, where we again observe small approximation errors across a wide range of perturbation magnitudes. Interestingly, the effect of the chosen perturbation magnitude on the approximation error is less pronounced than on MNIST, remaining low even for large perturbation magnitudes ($\epsilon = 0.01 > 8/255$). While the approximation error remains below 0.3% for a PGD-trained net, our approximation exhibits a consistent bias for the IBP-trained network, overestimating tightness by approximately 1.2%.

Figure 14: Mean relative error between local tightness (Definition 3.6) and true tightness computed with MILP for a `CNN3` trained with PGD or IBP at $\epsilon = 0.005$ on CIFAR-10.

### D.4 COMPARING TIGHTNESS TO (INVERSE) ROBUST CROSS-ENTROPY LOSS

To investigate to what extent our novel tightness metric is complimentary to the (inverse) robust cross-entropy loss (see Section 2) computed with IBP, we repeat the key experiments confirming our theoretical insights with the inverse IBP-loss and observe significantly different, partially opposite trends.

**IBP Loss at Initialization** We repeat our experiments on the dependence of tightness at initialization on network depth and width, illustrated in Figure 5, and additionally report the inverse IBP loss in Figure 15. For all experiments, we use the initialization of Shi et al. (2021) which has become the de-facto standard for IBP-based training methods. While we (theoretically and empirically) observe an exponential reduction in tightness with increasing depth, the inverse IBP loss increases slightly. Similarly, while we (theoretically and empirically) observe a polynomial reduction in tightness with increasing width, the inverse IBP loss stays almost constant. Note the logarithmic scale (and orders of magnitude larger changed) for tightness and the linear scale for the inverse IBP loss. This difference in trend is unsurprising as the custom initialization of Shi et al. (2021) is designed to keep IBP bound width constant over network depth and width. We thus conclude that tightness and (inverse) IBP loss yield fundamentally different results and insights when analyzing networks at initialization.

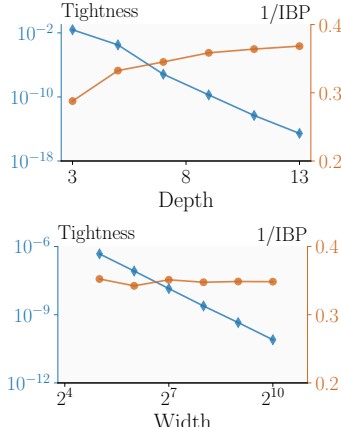

Figure 15: Tightness and inverse IBP loss at initialization depending on width and depth.

**IBP Loss after Training** We show the inverse IBP loss (left and center) and tightness (right) after IBP training depending on training perturbation size $\epsilon$, evaluated at training $\epsilon$ (left) or constant $\epsilon = 10^{-3}$ (center) in Figure 16. We observe that inverse IBP loss, in contrast to tightness, is heavily dependent on the perturbation magnitude used for evaluation (compare left and center), making it poorly suited to analyze the effects of changing perturbation magnitude. Further, when using the most natural perturbation magnitude, the $\epsilon$ used during training and certification (left), we observe completely different trends to tightness. For very small perturbation magnitudes, the inverse IBP

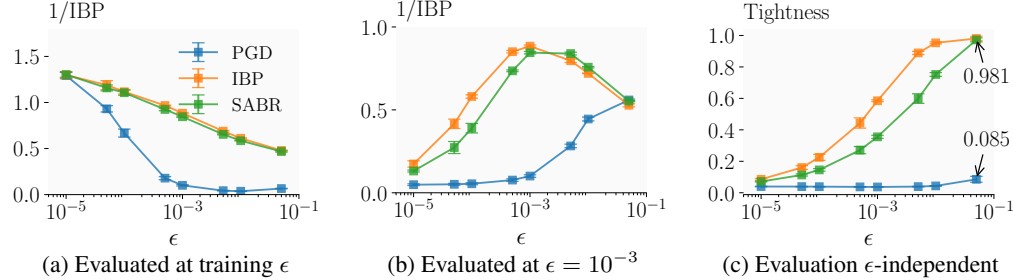

(a) Evaluated at training $\epsilon$     (b) Evaluated at $\epsilon = 10^{-3}$     (c) Evaluation $\epsilon$-independent

Figure 16: Tightness (right) and inverse IBP loss (left and center) after IBP training depending on training perturbation size $\epsilon$, evaluated at training $\epsilon$ (left) or constant $\epsilon = 10^{-3}$ (center).

Table 3: Certified and standard accuracy depending on network width.

| Dataset | $\epsilon$ | Method | Width | Accuracy | Certified | Tightness |
|---------|-----------|--------|-------|----------|-----------|-----------|
| MNIST | 0.1 | IBP | 1× | 85.70 | 67.71 | 0.871 |
| | | | 2× | 88.42 | 73.77 | 0.857 |
| | | | 4× | **90.31** | **79.89** | 0.803 |

loss is very high, suggesting high (perturbation) robustness, but both inverse IBP loss evaluated with a larger $\epsilon$ and tightness are low, showing that it neither permits precise analysis with IBP nor is necessarily robust, again highlighting the difference between tightness and (inverse) IBP loss.

### D.5 TIGHTNESS AFTER IBP TRAINING

To confirm that wider models improve certified accuracy while slightly reducing tightness across network architectures, we also consider fully connected networks, which used to be the default in neural network verification (Singh et al., 2019b; 2018). We increase the width of a fully connected ReLU network with 6 hidden layers from 100 to 400 neurons and indeed observe a significant increase in certified accuracy (see Table 3).

