# OpenReview forum: "Understanding Certified Training with Interval Bound Propagation"
_ICLR.cc/2024/Conference — ICLR 2024 poster_

### Official Review · Reviewer_KaHF · 2023-10-29

**Soundness:** 2 fair
**Presentation:** 3 good
**Contribution:** 1 poor
**Rating:** 5
**Confidence:** 5

**Summary:**

The paper introduces a notion of IBP tightness, defined as the ratio between the optimal output interval bounds and those obtained via IBP.
A series of technical results, mostly on Deep Linear Networks (DLN) is presented, describing the requirements for perfect tightness ("propagation invariance") and the influence of width, depth and IBP training on tightness.
Experimental results supporting the technical results are shown.

**Strengths:**

The research goal, providing further understanding to certified training and on the role of IBP in state-of-the-art methods, is of great interest to the community.
The paper is well-written and technical results are relatively coherent and well-structured.
The results detailing conditions for propagation invariance are novel and of potential interest to the community.

**Weaknesses:**

While the motivation is great, I am not sure this work is an actual step forward in understanding certified training.

Most of the results are either fairly obvious within the certified training community (at initialisation tightness will increase with width and depth) or follow from relevant and non-cited previous work on adversarial robustness [1] (for trained networks, width will help but depth won't).
The authors repeatedly overclaim: I do not see why this work would "pave the way for further advances in certified training", or "explain the success of recent certified training methods". For instance, concerning the latter, the SABR paper already shows quite clearly (in its Figure 7) that strong IBP regularization is not required to obtains state-of-the-art results.
I am also not fully convinced by the repeated arguments being made about the relevance of DLNs in this context. As it is also clear in the paper, if a network was locally linear in a perturbation region, any linear relaxation would imply exact certification. This is clearly extremely far from the networks that yield state-of-the-art certified training results, which typically require the use of branch-and-bound post-training to reach high certified accuracy (e.g., SABR).
Finally, the presented improvements over literature results (table 1) are somewhat overstated: for SABR, the best-performing methods amongst the ones considered, they are not even necessarily a strict improvement on both standard and certified accuracy (MNIST 0.3) and they come at the cost of memory and runtime overhead. Would these result hold on different perturbation magnitudes, for instance MNIST 0.1 and CIFAR-10 8/255?

[1] Robustness in deep learning: The good (width), the bad (depth), and the ugly (initialization), Zhu et al., NeurIPS 2022

**Questions:**

The main question I had throughout the paper is: why would this notion of tightness be any more useful than measuring the standard IBP loss of a network? Tightness does not necessarily imply robustness, as IBP bounds could be very close to optimality but the network may still display adversarial examples. On the other hand, a lower IBP loss will imply robustness. For instance, the custom initialization by (Shi et al., 2021) clearly improves certified training performance (as it brings down the IBP loss at initialisation) but this is not captured by the technical results on tigthness (as stated after corollary 3.8).
Furthermore, I am under the impression that most of the plots of the paper about tightness would apply to the (inverse of) the IBP loss too: decrease with width, depth, increase with IBP training and so on.
I believe this should be addressed and made an integral part of the paper.

---

> ### Author Response · Authors · 2023-11-17
> **Response to Reviewer KaHF Part I**
>
> We thank the reviewer for their interesting questions, which we address below, and are happy to hear that they believe both the problem we work on as well as our novel results to be of interest to the community.
>
>
> **Q1. Given that tightness does not directly imply robustness, why is the notion of tightness, introduced in this work, useful beyond the standard IBP loss? Further, does it yield any different trends than (inverse) IBP loss?**
>
> As this question is the Reviewer’s main concern, we break it down into multiple sub-questions and address them individually:
>
> *Q1.1: Does a low IBP loss imply robustness?*
> Not necessarily. For example, if the minimal logit difference to the runner-up class is 0, we can obtain a robust cross-entropy loss of $log(2)=0.7$, which is lower than the losses typically achieved with IBP training [1]. Similarly, as we scale the final layer weights towards 0, the IBP loss converges to $log(K)$, which is less than the IBP loss at initialization (see the new Figure 15) but will hurt certified training. Finally, networks trained with SABR often have a 10x larger IBP loss than IBP-trained networks, while actually being more robust (see Figure 10 in Müller et al. [1]). We thus conclude that a small IBP loss is neither necessary nor sufficient for robustness, and state-of-the-art SABR-trained networks in fact exhibit very high IBP losses.
>
> *Q1.2: Does the initialization of Shi et al., 2021 [2] improve certified training performance by reducing IBP loss (at initialization)?*
> We want to highlight that while Shi et al.’s initialization reduces IBP loss at initialization (although the first epoch (for CIFAR-10) does not use an IBP loss), they mostly prevent gradient explosions during the epsilon annealing phase of the training schedule thus allowing for much faster annealing and training schedules (see their Section 3.2.1). In fact, just using their initialization without additional regularizers can even hurt final performance (see their Table 4).
>
> *Q1.3: Do tightness and inverse IBP loss exhibit similar trends with width, depth, and perturbation magnitude?*
> No. In the new Figure 15 in Appendix D.4, we compare tightness and inverse BP loss over different depths and widths at initialization (using the de facto standard initialization by Shi et al. [2]) and observe opposite trends. The inverse IBP loss increases slightly with depth and stays roughly constant with width, while tightness decreases by multiple orders of magnitude in both cases. This is generally expected, as Shi et al.’s [2] initialization aims to keep the box size constant throughout the network. We have further computed the IBP loss for the settings and networks considered in Figure 7 using either a constant or the same epsilon as for training and evaluating the certified accuracy (Figure 16). We find the inverse IBP loss depends strongly on the epsilon used for evaluation. When using the same epsilon as for training (which we believe to be the most natural choice), we observe the largest inverse IBP losses for the smallest perturbation magnitudes, decreasing as perturbation magnitudes increase. This is the exact opposite of what we observe for tightness.
> In summary, inverse IBP loss shows completely different trends over architecture choices and (training) perturbation magnitudes. We have thus established the complementarity of tightness and the (inverse) IBP loss.
>
> *Q1.4: What is the value of the tightness metric as an analysis tool?*
> Tightness is the (to the best of our knowledge) first metric to consider the ratio of optimal and relaxed bound sizes in neural network verification. This allows us to directly quantify how strongly regularized a network is (or would need to be) for (IBP) certifiability and what methodological choices dominate this regularization strength. In contrast to the IBP loss, tightness disentangles robustness from accuracy. This is of particular importance as regularization for robustness, in particular higher tightness, often reduces accuracy, making their interaction hard to analyze.
> This perspective is, e.g., key to the interpretation of the results in Figure 5, suggested by Theorem 3.10, and leads to our architectural modifications that yield state-of-the-art results (Table 1). See also our answer to Q2 for concrete insights into, e.g., the mechanisms behind SABR training.
>
> **References:**
> [1] Müller et al. "Certified training: Small boxes are all you need." ICLR 2023
> [2] Shi et al. "Fast certified robust training with short warmup." NeurIPS 2021

---

> > ### Author Response · Authors · 2023-11-17
> > **Response to Reviewer KaHF Part II**
> >
> > **Q2. Can you discuss to what extent this work "explain[s] the success of recent certified training methods" as this seems like an overclaim? In particular, what do you add beyond Figure 7 of Müller et al. [1]?**
> > We first note that we in fact only claim to “**help** explain the success” of SABR. And indeed, we believe to provide interesting new insights:
> > Figure 7 of Müller et al. [1] shows that training with smaller propagation regions reduces regularization, which improves standard and adversarial accuracy and can improve certified accuracy if a sufficiently precise verification method is chosen. However, it remains unclear how the smaller propagation regions actually affect robustness and regularization, as they can still be chosen from the whole perturbation region. In our analysis in Figure 9, we observe that the regularization induced by SABR training is dominated by the propagation region size instead of the perturbation magnitude and in fact weaker for larger perturbation regions. This highlights the explicit trade-off between (robust) accuracy and regularization in the capacity-limited setting.
> > It also helps explain the much smaller improvements ($0.2\\%$ vs $6.4\\%$) SABR makes at high perturbation magnitudes  ($\epsilon = 8/255$ vs $\epsilon = 2/255$) where larger propagation regions ($\lambda=0.7$ vs $\lambda=0.1$) are required and thus tightness is reduced much less ($2.8\\%$ vs $77\\%$), see Table 2.
> >
> >
> > **Q3. Can you discuss to what extent most results are either fairly obvious (increasing tightness with width and depth at initialization) or follow from prior work [3]?**
> >
> > To avoid confusion, we want to highlight that, at initialization, tightness *decreases* rather than increases with width and depth. While an increase of *abstraction size* with width and depth is very intuitive (unless one used Shi et al.’s [2] initialization designed to prevent it) we are not aware of any other work analyzing the *abstraction precision*, i.e., the ratio of optimal vs relaxed bounds, thus we believe no corresponding result can be broadly dismissed as “obvious”. Further, even if the direction of the observed trends is not particularly surprising, we believe their precise asymptotics might be.
> > Additionally, we want to note that many retrospectively “obvious” results took multiple years to discover, e.g., Shi et al.’s [2] initialization scaling to keep abstraction sizes constant or Müller et al.’s [1] observation that (in contrast to prior assumptions) abstraction growth depends on input region size, are both very recent results.
> > Finally, we want to note that Zhu et al.’s [3] work considers a completely different setting from ours, and thus does not enable any direct conclusion relating to our results. While we consider certified (worst case) robustness, they consider average perturbation stability approximated using only the gradient at the original input. In particular, their results consider neither the worst-case aspect of certified robustness nor the effect of Interval Bound Propagation, which are the defining aspects of our work. We are happy to reference their results in our related work section, but again want to highlight that they do not permit any inference whatsoever regarding (IBP)-certified robustness, which we consider.
> >
> >
> > **Q4. How relevant are DLNs for the analysis of certified robustness given that if a network were locally linear for a full perturbation region, any linear relaxation would allow exact certification, which is not possible for state-of-the-art networks, requiring BaB for certification?**
> > We agree with the reviewer that state-of-the-art robust neural networks are not linear over the considered perturbation regions and never argue this to be the case. However, as with most theoretical work, we have to make carefully chosen simplifications to derive interesting and relevant results. As we observe only minimal errors (less than $0.5\\%$ at perturbation magnitudes equal to those used in training) in local tightness when comparing the exact tightness computed using MILP with our efficient approximation (see Figure 2 and the new Figure 13), we believe our insights to be relevant and applicable to ReLU networks. We further highlight that we extend some of our theoretical results to ReLU networks (see Corollary A.2).
> >
> > **References:**
> > [1] Müller et al. "Certified training: Small boxes are all you need." ICLR 2023
> > [2] Shi et al. "Fast certified robust training with short warmup." NeurIPS 2021
> > [3] Robustness in deep learning: The good (width), the bad (depth), and the ugly (initialization), Zhu et al., NeurIPS 2022
> > [4] Zhang, Huan, et al. "Towards stable and efficient training of verifiably robust neural networks." ICLR 2020
> > [5] Brix, et al. "First three years of the international verification of neural networks competition (VNN-COMP)." STTT 2023
> > [6] Müller et al. "Scaling polyhedral neural network verification on GPUs." MLSys 2021

---

> > > ### Author Response · Authors · 2023-11-17
> > > **Response to Reviewer KaHF Part III**
> > >
> > > **Q5. How significant are the reported improvements (Table 1) given that they are not always strict (MNIST 0.3) and come at the cost of memory and runtime overhead? Can you confirm them for MNIST 0.1 and CIFAR-10 8/255?**
> > >
> > > Improving certified robustness has proven to be an exceptionally hard problem. For example, on MNIST at $\epsilon = 0.3$, the state-of-the-art has only improved by $0.42\\%$ in the 3 years between CROWN-IBP [4] and SABR [1]. Thus, our improvement of $0.47\\%$ resulting from our theoretical insights exceeds that of 3 years of methodological research. Similarly, strict improvements are relatively rare in certified training (see Figure 6 or Figure 6 in Müller et al. [1]). Still, we observe strict improvements in 4 out of the 5 originally considered settings.
> > > We have now added experiments for MNIST at $\epsilon = 0.1$ and CIFAR-10 $\epsilon = 8/255$, again observing strict improvements in 3 out of 4 settings. We are additionally conducting experiments on TinyImageNet, where we observe strictly worse performance on narrower networks and are awaiting results on wider networks.
> > >
> > > Given that all (deterministically) certifiably robust models are relatively small (e.g., here a CNN with only 7 affine layers) and most computational cost is incurred at certification and not inference time, we believe any such improvements to be well worth the memory and runtime overhead. As recent precise verification methods require up to hundreds of seconds to verify a single input [5], while efficient but less precise methods, published multiple years ago, only require milliseconds [6], we believe this opinion to be shared by most of the field.
> > >
> > > We hope to have been able to address the Reviewer’s misconceptions and concerns, are happy to answer any follow-up questions, and look forward to their response.
> > >
> > > **References:**
> > > [1] Müller et al. "Certified training: Small boxes are all you need." ICLR 2023
> > > [4] Zhang, Huan, et al. "Towards stable and efficient training of verifiably robust neural networks." ICLR 2020
> > > [5] Brix, et al. "First three years of the international verification of neural networks competition (VNN-COMP)." STTT 2023
> > > [6] Müller et al. "Scaling polyhedral neural network verification on GPUs." MLSys 2021

---

> > > > ### Comment · Reviewer_KaHF · 2023-12-04
> > > > **I acknowledge the authors' detailed response and increase my score to 5. However I still cannot support acceptance.**
> > > >
> > > > I greatly appreciate the effort made by the authors in their response, and thank them for their time and clarifications.
> > > > In particular, I am happy that they analyzed the relationship to the inverse IBP loss, and appreciate that tightness conveys different information. What I am not fully convinced about, though, is its relative utility.
> > > > As the authors themselves state, Figure 15 is quite unsurprising owing to the use of the initialization from (Shi et al. 2021). It would be more interesting to see the results of the same experiment without that initialization, or post-training. I am also not fully convinced that Figure 16 is very informative, given that, as the authors state, the IBP loss will depend on the perturbation radius.
> > > > Nevertheless, in order to acknowledge the effort by the authors, I raised my score to 5.
> > > >
> > > >
> > > > Unfortunately, however, most of my concerns about the papers remain, and I cannot recommend acceptance. In this regard, I am happy that these were echoed by reviewer C4Wz. In particular, I am not convinced that, as the authors claim in the submission and the response, their empirical results follow from the provided theory.
> > > > Most of the recent certified training literature focuses on the same 7-layer network, and it is unsurprising that some architecture search (coupled with additional capacity) will improve performance.
> > > > Especially if the architectural changes are as simple as increasing a network's width or depth.
> > > > In addition, given that post-training verification relies on branch-and-bound, I still do not see why the empirical findings would directly follow from a study on the optimality of IBP bounds.
> > > > Some individual comments follow.
> > > >
> > > > >  We have now added experiments for MNIST at $\epsilon=0.1$ and CIFAR-10 $\epsilon=8/255$, again observing strict improvements in 3 out of 4 settings.
> > > >
> > > > In both cases, the reported SABR results for 1x are worse than those in the original paper. On CIFAR-10 $\epsilon=8/255$, the reported results for SABR with 2x are strictly worse than the original SABR results.
> > > >
> > > > > Most computational cost is incurred at certification
> > > >
> > > > Appendix C of SABR reports that, on TinyImagenet, training takes 2.5 days while certification takes 2 hours.

---

### Official Review · Reviewer_C4Wz · 2023-10-30

**Soundness:** 3 good
**Presentation:** 4 excellent
**Contribution:** 2 fair
**Rating:** 5
**Confidence:** 5

**Summary:**

This paper studies the interval bound propagation-based (IBP-based) training method, one of the most popular approaches to obtaining neural networks with certifiable guarantees. Although existing work demonstrated the effectiveness of this approach, the theoretical understanding of IBP is limited. Under the assumption of linear neural networks and Gaussian weights, this paper proposed a new measure of bound tightness and derived a few theorems to show how the bound tightness changes when propagating among linear layers, and how the width and depths of a randomly initialized network impact bound tightness.

Some empirical results on a few MNIST and CIFAR-10 models demonstrate that certified training indeed improves the tightness measure proposed in this paper. Some interesting empirical results were demonstrated with models varying depths and widths, demonstrating their correlations with bound tightness and accuracy.

The paper studies an important topic with some novel results, however, its current version has some weaknesses and unresolved questions, see below, so I feel the current version of the paper is below the acceptance threshold.

**Strengths:**

1. The topic of the paper is relevant, and it is an open challenge. We still don’t understand certified training very well, and this paper is a great attempt to bring in some new understanding.

2. Some novel theoretical insights are given, such as on the tightness of bound propagation and propagation invariance. Also, the growth of the bounds under initialization and its relation with model width may be a useful result to guide practical training.

3. The bound propagation invariance condition may lead to new insights into the design of neural network architecture to make the bounds tighter.

4. Some results are extended to a 2-layer ReLU network, although this part was not emphasized in the paper.

**Weaknesses:**

1. The theoretical results have strong assumptions such as linear neural networks, and neural network weights under Gaussian distribution. This is generally not a big concern if the authors can demonstrate that these theoretical insights can lead to great practical improvements,

2. but here the theoretical results developed do not lead to a better model that can outperform existing approaches, and some evaluations are quite weak (e.g., on a single MNIST model only). Since only a few models and networks are shown, it is unsure how general the results are.

3. Although some empirical results are shown to support the theory for ReLU networks, it is hard to argue these observations are indeed the consequence of the theory. For example, “certified training increases tightness” and “larger networks lead to better accuracy” are very generic conclusions, and it is hard to convince the readers that they result from the theory developed in this work.

**Questions:**

1. Figure 6 shows that increasing model width is beneficial, however, it is on a simple MNIST network. Can you demonstrate this result on larger networks and datasets? In particular, if we use a state-of-the-art method and model, and enlarge the model size by 4x, how much gain can we obtain over the state-of-the-art? Is the gain consistent over multiple models (e.g., CNN, resnet), training methods (IBP, CROWN-IBP, SABR, MTL-IBP), and datasets (CIFAR10, CIFAR100, TinyImageNet)?

2. Based on Theorem 3.4, can we reparameterize the network such that the bounds are always tight, and the training process just needs to search from a subspace of weights that lead to tight bounds, rather than using gradient descent to enforce tight bounds? For L2 norm certified defense, the state-of-the-art methods use this approach (such as orthogonal convolution layers and their variants).

---

> ### Author Response · Authors · 2023-11-17
> **Response to Reviewer C4Wz Part I**
>
> We thank the reviewer for their interesting questions and thoughtful suggestions. We are encouraged to hear that they consider our work to be both relevant and novel. Below, we address their remaining questions.
>
> **Q1. Can you discuss the implications of the (strong) assumptions (DLNs and Gaussian weight distributions) made during the theoretical analysis?**
>
> First, we want to highlight that the definitions of propagation invariance and bound tightness (Lemma 3.3 and Definition 3.6) do not assume a Gaussian weight distribution. Only our results on tightness at initialization make this assumption. There, however, it agrees well with commonly used initialization schemes, which draw weights from a Gaussian distribution.
> Second, while we do assume (local) linearity for many of our theoretical results, we want to highlight that ReLU networks are locally linear and thus behave linearly for infinitesimally small perturbation magnitudes. Even at the typically considered perturbation magnitudes, as few as 1% of ReLUs are unstable, leading to empirically small approximation errors (see Figure 2 and the new Figure 13). As we empirically validate all our results on ReLU networks and even generalize some of them theoretically to this setting (see Corollary A.2), we believe the insights obtained under the assumption of local linearity to be valuable and likely to generalize.
>
> **Q2. Can you discuss the generality of the empirical support of your theoretical findings given that some experiments are conducted for only one dataset and network architecture?**
>
> As most of our experiments involve sweeps over either architectures or perturbation magnitudes and certification with a precise neural network verifier, they can be quite compute-intensive. We have thus focused on what we believe to be the most informative and interesting settings, using the popular CNN7 (which yields state-of-the-art results in most settings) when trying to demonstrate performance improvements and a smaller CNN3 when computing exact bounds using MILP or aiming to certify PGD trained networks, which would be intractable for larger architectures.
> In addition, we confirm key results across multiple training methods and datasets (see below), e.g., while the results in Figure 6 already constitute a new state-of-the-art, we confirm it on CIFAR-10 in Table 1, where we also observe strict improvements on multiple state-of-the-art training methods. We have added new results (Figures 13 and 14) to the appendix confirming the results previously established in only one setting (Figures 2 and 9).
> Below, we provide an overview of results and settings we considered.
>
> If the reviewer believes any results to still be insufficiently established, we are more than happy to conduct additional experiments to address this.
>
> *Approximation error between local (Definition 3.6) and true tightness:* CNN3 on MNIST for 5 perturbation magnitudes (Figure 2). Now additionally for CIFAR-10 (Figure 13)
>
> *Tightness at initialization (Lemma 3.7 and Corollary 3.8, rigorous theoretical results):* 11 architectures based on CNN7 on CIFAR-10, although this is mostly dataset and completely perturbation magnitude independent, as we consider the network at initializtion (Figure 3).
>
> *Low dimensional embedding and reconstruction (Theorem 3.10, rigorous theoretical results):* 19 toy datasets constructed from projections of multivariate standard Gaussians (Figure 4).
>
> *Effect of depth on trained networks:* 6 architectures derived from CNN7 for CIFAR-10 at $\epsilon=2/255$ (Figure 5).
>
> *Effect of width on trained networks:* 6 architectures derived from CNN7 for CIFAR-10 at $\epsilon=2/255$ (Figure 5) and 3 training methods, two width factors, and 2 datasets (all using state-of-the-art networks as baseline) (Table 1 and Figure 6), now extended to 3 datasets and 5 perturbation magnitudes.
>
> *Effect of perturbation magnitude on trained networks (Theorem 3.9):* 8 perturbation magnitudes and 3 training methods for CNN3 and CIFAR-10 (Figure 7). 2 Perturbation magnitudes and 5 training methods for a state-of-the-art architecture from prior work (Table 2).
>
> *Regularization strength at same perturbation magnitude:* 2 training methods with 4 regularization strengths each for CNN7 on CIFAR-10 (Figures 8 and 12).
>
> *Perturbation magnitude vs propagation region size:* SABR training with 3 different lambda and 6 perturbation region sizes each for a CNN3 on CIFAR-10 (Figure 9), now also for MNIST (Figure 14).
>
> **References:**
> [1] Müller et al. "Certified training: Small boxes are all you need." ICLR 2023
> [2] Shi et al. "Fast certified robust training with short warmup." NeurIPS 2021
> [3] Mao et al. "Connecting Certified and Adversarial Training." NeurIPS 2023
> [4] Singh et al. "An abstract domain for certifying neural networks." POPL 2019

---

> > ### Author Response · Authors · 2023-11-17
> > **Response to Reviewer C4Wz Part II**
> >
> > **Q3. Can you demonstrate that the improved model performance with increased width observed for MNIST in Figure 6 also holds for other data sets, (state-of-the-art) architectures, and training methods?**
> >
> > *Settings:*
> > First, we want to note, that MNIST (at $\epsilon \in \\{0.1, 0.3\\}$) and CIFAR-10 (at $\epsilon \in \\{2/255, 8/255\\}$) are the four most frequently studied settings in certified adversarial robustness [1,2,3]. Between Figure 6 and Table 1, we considered two of these settings and show improvements in the state-of-the-art for both. We added experiments for the remaining two settings where we observed strict improvements in 3 out of 4 cases (see updated Table 1). Additionally, we are conducting experiments TinyImageNet and will add these results once completed.
> >
> > *Architectures:*
> > We consider a range of convolutional architectures derived from CNN7, which is used by all recent methods to achieve state-of-the-art results [1,2,3]. ResNets are not typically studied in the field, as they yield worse performance. We have confirmed these results for fully connected networks, which used to be the standard in the field [4], in Table 3, but believe the results around CNN7 to be of greater significance.
> >
> > *Methods:*
> > We study IBP as a baseline and two state-of-the-art methods, SABR [1] and IBP-R [5], which are both based on IBP. As we see improvements in certified accuracy across almost all methods and settings, we believe our findings to generalize across IBP-based methods.
> > We did not consider CROWN-IBP [7] as it is outdated, yielding worse results than IBP [2], and MTL-IBP [8] as its code has not been released.
> >
> >
> > **Q5. Can you discuss to what extent your empirical observations directly confirm your theoretical results?**
> >
> > While some of our key results can be easily confirmed experimentally, others are indeed more difficult to disentangle. We want to highlight that we confirm our results on tightness at initialization both for DLNs and ReLU networks in Figure 3. There, we show that tightness falls polynomially with increasing width (Lemma 3.7) and exponentially with increasing depth (Corollary 3.8). We further show that ReLU networks of fixed depth yield a constant factor increase in tightness (Corollary A.2). Similarly, we confirm the linear growth of the reconstruction error under IBP and square root growth under optimal propagation in the dimensionality reduction setting (Theorem 3.10 and Figure 4). While we can not rigorously show that the gradient alignment described in Theorem 3.9 is directly responsible for increasing tightness with IBP training, our empirical results (Figure 7 and Table 2) show that this increase in tightness is consistently observed for IBP-based methods, but to a much smaller degree (COLT) or not at all (PGD) for non-IBP based methods, even if they achieve good certified accuracy (COLT). Additionally, we observe that for SABR, tightness is dominated by the propagation region size and not the perturbation magnitude, agreeing well with Theorem 3.9 (Figures 9 and 14). We believe that taken together, these empirical results provide strong support for the applicability of our theoretical results on DLNs (rigorously proven in Appendix B) to ReLU networks (as acknowledged by Reviewer HbFM).
> >
> > **Q6. Can we reparameterize networks based on Theorem 3.4 such that the bounds are always tight?**
> > Great question! The easiest way to ensure propagation invariance based on Lemma 3.3 / Theorem 3.4 is to use non-negative weights. However, while this is sufficient it is not necessary and an even stricter constraint than implied by Theorem 3.4. Empirically, it leads to poor performance already in the non-robust setting [8]. We have further experimented with propagation invariant initializations based on Lemma 3.3 but have found the propagation invariant nature to be unlearned in the early stages of training and enforcing it without overregularizing to be non-trivial. We believe such a reparameterization to be interesting an
> > direction and promising item for future work.
> >
> > We hope to have been able to address all of the Reviewer’s concerns, are happy to answer any follow up questions, and look forward to their response.
> >
> > **References:**
> > [1] Müller et al. "Certified training: Small boxes are all you need" ICLR 2023
> > [2] Shi et al. "Fast certified robust training with short warmup" NeurIPS 2021
> > [3] Mao et al. "Connecting Certified and Adversarial Training" NeurIPS 2023
> > [4] Singh et al. "An abstract domain for certifying neural networks" POPL 2019
> > [5] De Palma et al. "IBP regularization for verified adversarial robustness via branch-and-bound" arXiv 2022
> > [6] Zhang, Huan, et al. "Towards stable and efficient training of verifiably robust neural networks" ICLR 2020
> > [7] De Palma et al. "Expressive Losses for Verified Robustness via Convex Combinations" arXiv 2023
> > [8] Chorowski and Zurada “Learning understandable neural networks with nonnegative weight constraints” TNNLS 2014

---

> > > ### Comment · Reviewer_C4Wz · 2023-11-23
> > > **Thank you for the detailed response. Still having concerns about the paper and cannot support it.**
> > >
> > > Thank you for the detailed response. I greatly appreciate the newly added results on different epsilon and models.
> > >
> > > After reading the author's response and comments from other reviewers, my main concern is the same as reviewer KaHF:
> > >
> > > > Most of the results are either fairly obvious within the certified training community (at initialisation tightness will increase with width and depth) or follow from relevant and non-cited previous work on adversarial robustness [1] (for trained networks, width will help but depth won't). The authors repeatedly overclaim
> > >
> > > I've read the response carefully, but to be honest, I am still not fully convinced by the claims made in this paper. The theoretical results and the experiments are somewhat disconnected, and it is hard to argue whether the observations are due to theoretical analysis.
> > >
> > > I appreciate the great effort the authors have made during the response period, so I will not decrease my score. Unfortunately, this key issue is still unsolved, so I cannot support the acceptance of this paper.

---

### Official Review · Reviewer_HbFM · 2023-11-01

**Soundness:** 3 good
**Presentation:** 3 good
**Contribution:** 3 good
**Rating:** 6
**Confidence:** 3

**Summary:**

This work studies the certified training from the theoretical perspective and applies it to explain and understand the mechanism of certified training with IBP in robustness certification. The idea of propogation tightness is formulated to analyze how IBP works and extensive experiments validate the theories from different aspects, including network width and depth, tightness and accuracy, etc.

**Strengths:**

- The motivation of the work makes sense to me and the theory is sound, especially I like the formulation of tightness in terms of optimal box and layerwise box.
- The paper is generally well-written and easy to read, and there are some easy examples to help the audience follow.
- The experiments are comprehensive, which mostly validates the theory part and gives many interesting insights for certified training.

**Weaknesses:**

-  Although there are some examples in the introduction and formulation, the theory details lack some intuitive insights or explanations, e.g. Theorem 3.9 needs more insights to make it intuitive as it is one of the core theorems in this work.
- The details of the experiments are not given in the main text; specifically, the datasets and models used in Fig. 3 are not clear. It is better to re-organize experiments part by adding a setup subsection for these necessary details.
- Why is the certified accuracy decreasing when $\epsilon$ increases in the middle figure in Fig. 7? More explanations and justifications are needed. Besides, it seems that there is no explanation of the trade-off between accuarcy and robustness as claimed in the abstract and introduction, excpet for tightness defined in the work, how about the certified robustness (e.g. certified accuracy or certifed radii) for the trade-off?

**Questions:**

See weakness

---

> ### Author Response · Authors · 2023-11-17
> **Response to Reviewer HbFM**
>
> We thank the reviewer for their interesting questions and thoughtful suggestions. We are encouraged to hear that they consider our work to be well-motivated and our experiments comprehensive. Below, we address their remaining questions.
>
> **Q1. Can you add more intuitions for the theoretical results you prove, e.g., Theorem 3.9?**
> While many of the proofs are quite technical, the results are intuitive, we have thus focused on providing intuitions on the results rather than the proof techniques.
> For example, the proof of Theorem 3.9 uses a first-order Taylor approximation (in perturbation magnitude) of the robust risk to show that the gradient change $R(\epsilon + \Delta \epsilon) - R(\epsilon)$ with increasing $\epsilon$ is aligned with the tightness gradient. We then sum over many perturbation magnitude increments $\Delta \epsilon$, and use the linearity of the inner product to obtain that $R(\epsilon) - R(0)$ is also aligned with the tightness gradient. As we let $\Delta$ go against 0, we obtain our result.
> The best intuition we can give is that every increase of the perturbation magnitude makes the gradient more aligned with tightness.
>
> **Q2. Can you provide more details on the experimental setup? For example, the setting of Fig. 3 remains unclear.**
> Of course! We have added the key details for every experiment (in particular for Figures 2, 3, and 9) to the main text. However, as we consider a large number of training methods and multiple datasets, each with different (default for these methods) hyperparameters, we defer additional details to Appendix C (almost one page) due to space constraints.
>
> **Q3. Why is the certified accuracy decreasing as $\epsilon$ increases in the middle part of Figure 7?**
> In Figure 7, we vary the $\epsilon$ used for both training and testing. Thus, the certified accuracy in the middle of Figure 7 is w.r.t. larger perturbation magnitudes as $\epsilon$ increases, which makes it a strictly harder task. Thus two effects come together. First, standard accuracy decreases (Figure 7 left) as networks are more heavily regularized for these larger perturbation magnitudes (see increases in tightness in Figure 7 right). Second, for the same network, every sample that is robust at $\epsilon$ is also robust at $\epsilon’ < \epsilon$ but not vice versa, thus certified accuracy is always monotonically decreasing with perturbation magnitude (for a fixed network).
>
> **Q4: Can you provide a more detailed explanation of your results on the robustness accuracy trade-off and in particular how your results on tightness relate to certified accuracy?**
> Yes. We argue along the following lines: To achieve high certified accuracies, certified training is required*. Empirically, IBP-based training methods have proven the most successful, yielding state-of-the-art results in every setting [1,2,3]. However, training with IBP increases tightness (Theorem 3.9), which induces strong regularization (Theorem 3.4), which in turn leads to reduced accuracy. We thus provide an explanation for the robustness accuracy trade-off for IBP-based training methods, which are practically the most relevant group of certified training methods.
>
> *We note that while the combination of a relatively small network (CNN3) and a complete verifier (MN-BaB) yields comparatively high certified accuracies for the PGD-trained network in Figure 7, especially at small perturbation magnitudes, certified training methods are required for state-of-the-art performance (which require larger networks).
>
> We hope to have been able to address the reviewer's concerns, are happy to answer any follow-up questions, and look forward to their reply.
>
> **References:**
> [1] Müller et al. "Certified training: Small boxes are all you need." ICLR 2023
> [2] Mao et al. "Connecting Certified and Adversarial Training." NeurIPS 2023
> [3] De Palma et al. "Expressive Losses for Verified Robustness via Convex Combinations." arXiv 2023

---

### Official Review · Reviewer_FpVR · 2023-11-01

**Soundness:** 3 good
**Presentation:** 3 good
**Contribution:** 2 fair
**Rating:** 6
**Confidence:** 3

**Summary:**

This paper provides theoretical analysis on IBP training and helps explain the success of IBP training over other non-IBP methods.

**Strengths:**

1. The paper gives a definition of the global and local propagation tightness which is new in the literature.
2. Theorem 3.9 gives a pretty interesting result that IBP improves tightness by proving the alignment between gradients.

**Weaknesses:**

1. Some analysis in Section 4.1 is not clear, please see the questions below.
2. What can be the potential improvement for certified training methods from your analysis?
3. Some missing related works:
     - [1] has a relevant conclusion on the diminishing improvement with increasing width in IBP training.

[1] On the Convergence of Certified Robust Training with Interval Bound Propagation

**Questions:**

1. In figure 5, why does a decreasing tightness lead to higher accuracy? If a looser bound is better due to weaker regularization, why is increasing the depth worse than increasing the width?

---

> ### Author Response · Authors · 2023-11-17
> **Response to Reviewer FpvR**
>
> We thank the reviewer for their insightful questions, which we address below and are encouraged to hear that they acknowledge our results to be interesting and novel.
>
> **Q1. Why does a lower tightness lead to higher accuracy in Figure 5? And why is increasing depth worse than increasing the width if less tightness is desirable?**
> At initialization, both higher width and depth decrease tightness significantly (see Figure 3). Then, IBP-based training increases tightness (Theorem 3.9), inducing (strong) regularization in the process (Theorem 3.4). The larger this increase in tightness, the stronger the induced regularization. As (IBP-)trained networks exhibit roughly similar tightness across depths and widths (compared to those at initialization), smaller tightness at initialization requires stronger regularization to reach this level (see the updated Figure 5). If this increase in regularization dominates the increase in capacity, we achieve worse goodness of fit (training set robust accuracy) and thus performance. This is what we observe in Figure 5, tightness increases much more over training for large depth than for large width, indicating stronger regularization, explaining the worse certified accuracy. So less tightness is only a positive sign (of less regularization) after IBP-training.
>
> **Q2. Can you relate your results to those of Wang et al. [1] which conclude that there are diminishing improvements with increasing width in IBP training?**
> First, we want to highlight that we already cite Wang et al. [1] in our related work section. Their work considers the convergence of IBP training in the over-parameterized setting for small perturbation magnitudes with high probability and finds that sufficient width is required to achieve zero training loss with high probability. In this setting, the required width increases inversely proportional to the square of the non-convergence probability (for very small perturbation magnitudes). However, while this can be seen as diminishing “improvements” of convergence probability with increasing width, we observe diminishing improvements in test set certified accuracy, a fundamentally different metric. Further, the work of Wang et al. [1] is limited to 10x smaller perturbations than we consider (see e.g. their Figure 1).
>
> **Q3. Can you highlight the potential improvements to certified training enabled by your analysis?**
> We believe that tightness can become a valuable practical and theoretical analysis tool for designing and understanding new certified training methods, as it allows measuring the regularization strength induced by IBP-based training. For example, we demonstrate in Figure 9 that tightness and thus regularization strength are dominated by the size of the propagation region instead of the permissible perturbation magnitude, providing deep insights into the success of SABR [2].
> Further, it provides a promising new angle to theoretically investigate IBP-based training. Finally, our theoretical results directly predicted architectures yielding improvements over the previous state-of-the-art (see Table 1 and Figure 6).
>
> We hope to have been able to address all the reviewer's concerns, are happy to answer any follow-up questions they might have, and are looking forward to their reply.
>
> **References:**
> [1] Wang et al. "On the convergence of certified robust training with interval bound propagation." ICLR 2022
> [2] Müller et al. "Certified training: Small boxes are all you need." ICLR 2023

---

> > ### Comment · Reviewer_FpVR · 2023-11-22
> >
> > Thanks for the authors' feedback.
> > For the first question, if less tightness is better due to weaker regularization, can we just make the IBP box larger for better performance?

---

> > > ### Author Response · Authors · 2023-11-22
> > > **Thanks for engaging in the discussion and follow-up.**
> > >
> > > We thank the reviewer for engaging in the discussion!
> > >
> > > Indeed, reducing the resulting tightness by changing the propagation region size lies at the core of the state-of-the-art certified training method SABR (see the experiment illustrated in Figure 9 and the new Figure 14). However, we have to reduce the propagation region/box size to reduce regularisation and thus tightness during training.
> > >
> > > We further want to highlight that measuring local tightness (see Definition 3.6) is independent of any box size allowing us to analyze and isolate these effects (for the first time).

---

> > > > ### Comment · Reviewer_FpVR · 2023-11-22
> > > >
> > > > Thanks for the answer. I have increased the rating to 6 for now.
> > > > But the result still seems counter-intuitive for me. Following Def 3.6, if we control the same optimal box size, larger tightness means smaller actual box radius. Then this should actually leads to a weaker regularization as the difference between standard training and certified training with IBP is smaller.
> > > > Could the authors explain more on this?

---

> > > > > ### Author Response · Authors · 2023-11-22
> > > > > **Follow Up**
> > > > >
> > > > > We thank the reviewer for updating their score and are more than happy to provide an explanation:
> > > > >
> > > > > Tightness as per Def. 3.6 can be seen as a measure of regularization for precise Box propagation at the cost of accuracy. Crucially, we compare output box sizes which we can not influence directly, but only indirectly via regularization strength.
> > > > >
> > > > > Training with a larger propagation region/box size increases regularization strength and thus leads to smaller layer-wise propagated boxes compared to the optimal box size and thus increased tightness for the *trained network* (see Fig. 7). Conversely, training with a smaller propagation region/box size reduces regularization strength and thus leads to larger layer-wise propagated boxes and thus reduced tightness for the trained network (see again Fig. 7).
> > > > > Perhaps this was the reviewers suggestion all along and we misunderstood their comment.

---

### Author Response · Authors · 2023-11-17
**General Response**

We thank all reviewers for taking the time to provide insightful feedback and ask interesting questions. We are particularly encouraged to hear that reviewers consider our work to be well-motivated (*HbFM*), novel (*RpVR, C4Wz, KaHF*), interesting/relevant to the community (*RpVR, HbFM, KaHF, C4Wz*), well-written (*HbFM, KaHF*), and supported by a comprehensive empirical evaluation (HbFM).

As we have not identified any shared concerns, we address the reviewers’ questions in individual responses. We provide more detail on experimental setups (changes highlighted in blue) in the updated rebuttal revision and have added the following experiments:
* We confirm the importance of sufficient width for certified accuracy, in 3 additional settings for multiple training methods (updated Table 1): MNIST at $\epsilon = 0.1$, CIFAR-10 at $\epsilon = 8/255$, and TinyImageNet at $\epsilon = 1/255$. In 10 out of 11 settings, we achieve the best certified accuracy at the largest width with strict improvements in 9 of these. We want to highlight that for all results in Table 1, we consider the state-of-the-art architectures as baselines.
* We confirm the small approximation errors between local (Definition 3.6) and true tightness, previously only shown for MNIST (Figure 2), on CIFAR-10 (Figure 13).
* We confirm the dominance of the propagation region size for tightness, previously only shown for CIFAR-10 (Figure 9) on MNIST (Figure 14).
* We confirm our observations on the effect of increased width on certified accuracy and tightness from Table 1 and Figures 5 and 6 for fully connected architectures (Table 3).
* Comparing Inverse IBP loss and Tightness: We repeat the experiments analyzing the interaction of tightness and training perturbation magnitude (Figure 7 => Figure 16), and network width and depth at initialization (Figure 5 => Figure 15) for the inverse IBP loss in Appendix D4. We show that the IBP loss exhibits completely different trends and is complementary to tightness.

---

### Author Response · Authors · 2023-11-21
**Rebuttal Reminder**

As the rebuttal period is slowly drawing to a close, we would like to encourage the Reviewers to pose any follow-up questions they might have and let us know if we were unable to address all of their concerns.

In any case, we would greatly appreciate the Reviewers taking the time to acknowledge our rebuttal.

---

### Author Response · Authors · 2023-11-22
**Extended empirical evaluation**

Based on the Reviewers' suggestions, we have extended Table 1 to all settings commonly considered in certified adversarial robustness, including 5 different perturbation magnitudes across 3 different datasets (MNIST, CIFAR-10, and TinyImageNet) for 3 different training methods.
We confirm our original results and show that the wider architecture suggested by our theoretical insights yields strict improvements in certified and standard accuracy in 9 of the 11 considered settings, with improvements in one of the two metrics in the remaining two settings.

We hope this addresses the reviewers' concerns regarding the general applicability of our results.

---

### Meta-Review · Area_Chair_TMbz · 2023-12-08

**Metareview:**

The authors propose a novel metric for analyzing the quality of Interval Arithmetic based bounds (IBP) on the activations of a neural network. They demonstrate that this metric can be analyzed for simple networks like linear networks with random intialization, and that the metric correlates well with the performance of IBP based training procedures. Two reviewers raised several questions through the rebuttal phase, the primary one being whether the metric studied by the authors truly captures precisely what leads to the success of an IBP based certified training method. The authors engaged in the discussion well, and, in my opinion, provide reasonable albeit not comprehensive evidence of the validity of their results. While the reviewers remain unconvinced, I think the paper can be accepted while requesting that the authors revise their paper to incorporate the comments of the two reviewers who were unconvinced, and include shortcomings of their approach and avenues for further work.

**Justification For Why Not Higher Score:**

Empirical evidence does not fully justify the claim that the measure proposed by the authors accurately captures when certified training would work well.

**Justification For Why Not Lower Score:**

The paper contains interesting insights that could be valuable to the ICLR community working on certified robustness.

---

### Decision · Program_Chairs · 2024-01-16

Accept (poster)